# Modeling Adhesive Hysteresis

**Anle Wang, Yunong Zhou and Martin H. Müser \*** 

Department of Material Science and Engineering, Saarland University, Campus C6 3,
66123 Saarbrücken, Germany; anle.wang@uni-saarland.de (A.W.); yunong.zhou@uni-saarland.de (Y.Z.)
\* Correspondence: martin.mueser@mx.uni-saarland.de

**Abstract:** When an elastomer approaches or retracts from an adhesive indenter, the elastomer's surface can suddenly become unstable and reshape itself quasi-discontinuously, e.g., when small-scale asperities jump into or snap out of contact. Such dynamics lead to a hysteresis between approach and retraction. In this study, we quantify numerically and analytically the ensuing unavoidable energy loss for rigid indenters with flat, Hertzian and randomly rough profiles. The range of adhesion turns out to be central, in particular during the rarely modeled approach to contact. For example, negligible traction on approach but quite noticeable adhesion for nominal plane contacts hinges on the use of short-range adhesion. Central attention is paid to the design of cohesive-zone models for the efficient simulation of dynamical processes. Our study includes a Griffith's type analysis for the energy lost during fracture and regeneration of a flat interface. It reveals that the leading-order corrections of the energy loss are due to the finite-range adhesion scale at best, with the third root of the linear mesh size, while leading-order errors in the pull-off force disappear linearly.

**Keywords:** adhesion; cohesive zone model; hysteresis

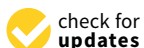



## 1. Introduction

Adhesion between solid bodies plays an important role in nature and technology. Usually, it is strongly suppressed due to the presence of roughness [1,2], which exists even for highly polished surfaces. However, when one of the two solid bodies is very compliant and both are smooth, adhesion can become noticeable at relatively large scales and be exploited technologically [3].

The optimization of adhesive structures can certainly benefit from modeling adhesion, which, however, is not always a trivial task. One difficulty is that adhesion tends to be very short ranged, which leads to stiff differential equations to be solved when describing a structure at a coarse scale. A popular method to avoid singularities and to reduce the stiffness of adhesive contact problem is to use so-called cohesive zone models (CZMs) [4–6]. They describe, usually in analytical form, how the traction depends on the local separation between two surfaces. CZMs are commonly stated and used for a given pair of surfaces irrespective of the scale to which the surface is discretized.

Traditionally, CZMs [7] are constructed in a top-down fashion, i.e., surface energy $\gamma$ and Tabor parameter $\mu_T$, a measure for the range of adhesion, are determined at an intermediate length scale, and the parameters of a given CZM are adjusted such that a desired pull-off stress is produced. It was shown for adhesive Hertzian contacts that details of the functional form of CZM's do not significantly affect how contact area and displacement and thus the pull-off stress change as a function of normal load, as long as $\gamma$ and $\mu_T$ are matched [8].

Only few attempts have been made so far to construct CZMs from the bottom up [9,10]. This includes the systematic elimination of small-scale random roughness features leading to a reduction of surface energies and an increase in the interaction range with respect to the interatomic potentials [11,12]. While the stiffness of the mathematical problem is certainly substantially alleviated by such coarse graining for most nominally plane

surfaces, the traction laws to be used may remain rather short ranged for adhesive system, for which random roughness does not substantially reduce the surface energy below the typical values of $\mathcal{O}(100)$ mJ/m$^2$. The desirable coarse of action may then remain to model adhesion as being as short-ranged as needed but as long-ranged as possible.

Using CZMs that reflect the microscopic short range of adhesion realistically either requires a fine discretization or induces unrealistic force-displacement dependencies [13]. When the grid is not sufficiently fine, jump-in or snap-out dynamics usually suffer from unacceptably large errors, e.g., the pull-off force and work of separation can be largely overestimated [14]. A frequent solution to this problem is a mesh refinement in the zone of interest, which, however, implies a low computational efficiency. Unfortunately, there does not appear to be a generally accepted or well tested rule for how to best select the mesh. When it cannot be made very fine, the most common way to proceed is to reduce the surface energy, whereby realistic traction forces [4,15–20] can be obtained. However, it is doubtful that this is the best course of action, since the simulation of dynamical processes requires the total energy balance to be accurate and not only the traction at the contact edges. Underestimating surface energy in the contact region, where the mesh is coarse, destroys that balance.

In this work, we propose a rule for how to select the mesh size for a given CZM, and more importantly, we provide a recipe for how to redesign it such that it provides accurate force-displacement dependencies if the mesh size cannot be made arbitrarily small. Towards this end, we focus on the case of a smooth flat elastomer in contact with a rigid, flat, smooth indenter with adhesive interaction as the most basic model. A central goal is to construct a CZM in which the energy hysteresis occurring in a compression-decompression loop are as accurate as possible. Much of the underlying analysis re-investigates the question how, or, rather when originally flat, soft-matter surfaces become unstable on approach to a flat adhesive counterface before making contact [21,22]. This in turn brings us to another issue, which has been discussed surprisingly little, namely, whether a CZM reproduces not only retraction but also approach realistically.

It is easily found, as in this contribution, that a Tabor parameter of $\mu_T = 1$ is sufficiently large to produce a load-displacement curve in contact similar to that obtained in the limit of infinitesimally short-range adhesion, which was solved by Johnson, Kendall, and Robertson (JKR) [23]. However, the approach is scarcely ever explored, although it is decisive for the unavoidable hysteresis that ensues as a consequence of the difference between the approach and the retraction curve. In a study addressing a full approach and retraction cycle of an adhesive Hertzian indenter, Ciavarella et al. [24] found that the energy loss is substantially reduced by finite Tabor parameters $\mu_T$ compared to the idealized case of zero-range adhesion, e.g., for $\mu_T = 5$ by almost a factor of two and by still $\mathcal{O}(20\%)$ for a Tabor parameter as large as $\mu_T = 50$. This latter value would be characteristic of a soft-matter interface with the following order-of-magnitude specifications: local radius of curvature of 100 nm, surface energy of 100 mJ/m$^2$, interaction range 4 Å, and a contact modulus $E^*$ of 10 MPa. Based on this ballpark estimate, it appears as if most soft-matter systems are locally sticky, even if typical surface roughness at large wavelength destroys that stickiness at macroscopic scales [11,25,26]. Rather than simulating such interfaces with $\mu_T = 50$, it might be desirable to run simulations with much smaller $\mu_T$ and to extrapolate to the desired $\mu_T$, which may often be approximated as infinity. While Ref. [24] certainly contains implicitly a recipe for this interpolation, it is neither explicitly stated nor is it clear if it extends to geometries beyond parabolic tips. Thus, one purpose of this work is to explore how to extrapolate numerical results for adhesive losses to short ranges.

The reminder of this paper is organized as follows: The model and the computational method are presented in Section 2. Section 3 contains analytical and numerical approaches to the contact between two adhesive, originally flat adhesive surfaces, including a guideline for the construction of scale-dependent CZMs. While none of the analytical results might be new, we obtain them from "first-principles" without making direct use of linear fracture

mechanics. The guidelines identified for smooth surfaces are then applied to uneven indenters in Section 4. Conclusions are drawn in the final Section 5.

## 2. Model and Method

### 2.1. Model

We consider an adhesive, flat, linearly elastic, semi-infinite elastomer interacting with a rigid indenter. The center-of-mass of the elastomer's bottom surface, $u_0$, is gradually decreased from a large positive value, clearly exceeding the characteristic length of attraction, to a value, where elastomer and indenter repel each other and then increased again back to its original value. The internal degrees of freedom, as denoted by $u(\mathbf{r})$ in real space or by its Fourier transform $\tilde{u}(\mathbf{q})$, are allowed to take arbitrary values except for the center-of-mass mode $u_0 = \tilde{u}(0)$, see Figure 1. The elastic energy to deform the (surface of the) elastomer is given by

$$V_{\text{ela}} = A \sum_{\mathbf{q}} \frac{q E^*}{4} |\tilde{u}(\mathbf{q})|^2, \tag{1}$$

where $E^*$ is the elastomer's contact modulus and $q$ is the magnitude of $\mathbf{q} = (q_x, q_y)$. The square domain has an area of $A = L^2$, where $L$ is the system's linear dimension. The central image is repeated periodically in $x$ and $y$ direction,

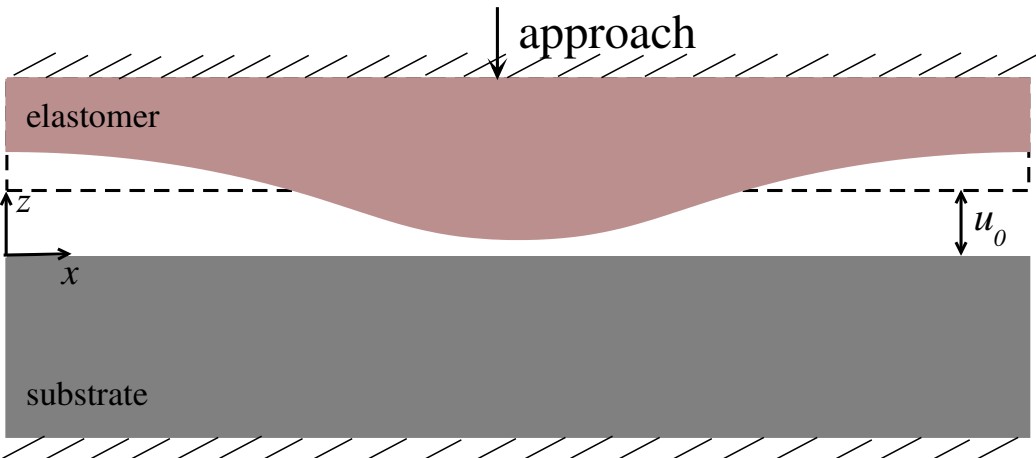

**Figure 1.** Schematic illustration of the computational model. The elastomer is moved relative to a rigid indenter such that the center-of-mass position of the elastomer's lower surface is constrained to a (time-dependent) value. The remaining internal degrees of freedom are allowed to relax to a configuration minimizing the total (potential) energy.

The default geometry of the rigid indenter is flat, however, uneven surfaces are considered as well. The $xy$-plane is located such that it cuts through the indenter's highest point. The contact between elastomer and indenter is frictionless. Furthermore, the interfacial energy per simulation cell is defined as

$$V_{\text{int}} = \int_A d^2r \, \gamma(\mathbf{r}) \tag{2}$$

with the interfacial energy density given by, for example, a relation inspired by the Morse potential

$$\gamma_{\text{M}}(\mathbf{r}) = \gamma \left[ e^{-2\{g(\mathbf{r}) - \rho_0\}/\rho} - 2 \, e^{-\{g(\mathbf{r}) - \rho_0\}/\rho} \right], \tag{3}$$

where $\gamma$ is the (maximum) surface energy, $\rho$ the decay length of the adhesion, and $\rho_0$ the equilibrium separation between indenter and elastomer. The latter is set to $\rho_0 = 0$, since it constitutes only an offset, which can be deemed irrelevant in a continuum treatment. The function $g(\mathbf{r}) = u(\mathbf{r}) - h(\mathbf{r})$ indicates the gap or interfacial separation between elastomer

and indenter as a function of the in-plane coordinate $\mathbf{r} = (x, y)$, where $h(\mathbf{r})$ states the shape of the indenter. For a flat indenter, $h(\mathbf{r}) \equiv 0$.

The original rationale for the choice of this particular interaction law, which is also known as Morse potential, was as follows: An exponential attraction as cohesive zone model was reported to yield smooth responses [27,28]. For reasons that should become obvious later in this work, we want the interaction to be at least twice differentiable so that a hard-wall repulsion is no option. The Morse potential is then beneficial, because the repulsive stress can be computed by squaring the exponential function $\exp\{-g(\mathbf{r})/\rho\}$ without having to evaluate another exponential. Moreover, the curvature in the energy minimum is relatively modest (which is beneficial for simulations). Finally, it is relatively easy to change the interaction range by replacing $\rho$ with a different value, without having to reparametrize the prefactor $\gamma$.

Alternatively, it would have been possible to use, for example, a $m - n$ Mie potential,

$$\gamma_{\mathrm{Mie}}(g) = \gamma \, \frac{m\,n}{m-n} \left\{ \frac{1}{m} \left( \frac{g}{\rho} \right)^{-m} - \frac{1}{n} \left( \frac{g}{\rho} \right)^{-n} \right\}$$

with $m > n > 0$ being real numbers. The Mie potential is sometimes misleadingly said to be a generalization of Lennard–Jones, however, Mie [29] introduced his potential more than two decades before Lennard–Jones [30]. An effective $8 - 2$ Mie potential between surface points ensues from Lennard–Jones interactions between two semi-infinite bodies within the Derjaguin approximation [31].

Since both considered potentials have the property that repulsion decreases more quickly with distance than attraction, they should lead to qualitatively similar behavior, just like other potentials with that property. However, moderate changes in the adhesion law can still affect some computed properties quite substantially. This is why some thought should be spent on the choice of the potential. If the goal is to construct a CZM starting from the atomic scale, a properly constructed Mie potential would be a good candidate, in particular if the adhesion arises mainly from dispersive or van-der-Waals forces. If, however, the mesh-elements are more than a few microns in size, the CZM should reflect the proper contact mechanics of the underlying microscopic (random) roughness and the functional form be chosen accordingly. As we find in preliminary simulations of adhesive, randomly rough surfaces, these CZMs can be similar to the Morse potential, as they can be well described by a difference between two exponentially decaying functions. In fact, a purely repulsive, non-overlap constraint between an elastomer and a randomly rough surface effectively leads to an exponential between the two surfaces [32,33]. If, however, the goal is to reach the continuum limit as quickly as possible, yet different choices are possible, e.g., the one introduced later in Equation (31).

For the simulations on ideally flat surfaces in this study, we decided to use the Morse potential. In hindsight, we could argue that this was done to represent the formation and the detachment of a randomly rough surfaces at a coarse scale. Two properties of the Morse surface-energy density are needed in the remainder of this article. First, the maximum tensile traction, i.e., the maximum of the first derivative of the r.h.s. of Equation (3). It is given by $\sigma_{\max} = \gamma/(2\rho)$ and located at an interfacial separation of $g = \rho \ln 2$. Second, the negative minimum curvature, which can be deduced to be $\kappa_{\max} = \gamma/(4\rho^2)$. It occurs at an interfacial separation of $g = \rho \ln 4$. Moreover, note that the radius of curvature of a flat contact is formally infinite (at least in the limit $L \to \infty$) so that the (usual) Tabor parameter can be said to diverge automatically and thus the interaction to be short-ranged irrespective of the numerical value of $\rho$.

## 2.2. Method

The system is displacement-driven rather than force-driven, i.e., depending on the mean gap $u_0$ between elastomer and indenter, the total potential energy

$$V_{\mathrm{tot}}[g_0, u(\mathbf{r})] = V_{\mathrm{ela}}[u(\mathbf{r})] + V_{\mathrm{int}}[g_0, u(\mathbf{r})] \tag{4}$$

is minimized by a structured or unstructured displacement field $u(\mathbf{r})$. Minimization is done using Green's function molecular dynamics (GFMD) [34], in which the elastomer is discretized into $(L/a_0) \times (L/a_0)$ square elements, $a_0$ being the linear discretization so that the number of grid points in $x$ and $y$ direction are identical $n_x = n_y = L/a_0$. The Fourier transforms $\tilde{u}(q)$ are used as the dynamical degrees of freedom. Here, we employ the so-called mass-weighting GFMD variant as described in Ref. [35], because of its high convergence rate. The basic idea of mass-weighting is to assign inertia to each $\tilde{u}(\mathbf{q})$ mode such that the system's intrinsic frequencies collapse as well as possible. This can be achieved by choosing the inertia roughly inversely proportional to $q$. The equations of motion were augmented with a thermostat as described in Ref. [36], in order to introduce small, symmetry breaking perturbations to the displacement field. The thermal noise induces a quicker transition from an unstructured displacement field, $u(\mathbf{r}) \equiv$ const, to a structured one than round-off errors. The thermal energy is chosen to be very small so that it does not significantly assist the elastomer to overcome energy barriers. It is yet large enough to make the elastomer quickly "realize" when a displacement field is no longer stable against a small perturbation.

The mean gap, or in the case of Hertzian indenter, simply the displacement, is moved quasi-continuously using a ramp, which in most cases, was realized as follows: For 50 time steps, $u_0$ is changed over a small quantum $\Delta u_0$. The system is then relaxed over typically 150 additional time steps. In most cases, this is sufficient to closely approach the next stable or metastable configuration. For a $512 \times 512$ system, one increment in average displacement then takes a little less than 1.5 s using our house-written GFMD code on a single core of a 1.6 GHz Intel Core i5 processor. For larger systems, the number of necessary time steps to be done per $\Delta u_0$ does not increase with system size due to the mass-weighting procedure.

## 3. Patterns and Instabilities in Periodically Repeated, Flat, Adhesive Contacts

Adhesion is known to lead to instabilities when two surfaces approach each other. The arguably simplest description of an adhesive instability was proposed by Tomlinson [37], who assumed atoms to be bonded to their lattice sites by springs of stiffness $k$. As a surface atom approaches a counterface, the position of the atom becomes unstable when the negative curvature of the atom-surface interaction exceeds $k$, in which case, the atom jumps into contact. On retraction, the inverse jump occurs at an increased separation between the equilibrium site and the counter surface, so that hysteresis and thus energy dissipation results.

It is now well known that Tomlinson's model is not sufficiently refined to describe adhesive hysteresis. Its simplest valid description was proposed by Johnson, Kendall, and Robertson (JKR) [23]. In their solution of short-range adhesion in Hertzian contact geometries, jump to contact occurs at a zero load, but breaking the same contact on retraction requires the tensile load and the work of adhesion to be finite.

In a tribological context, surprisingly little attention has been paid to flat, adhesive interfaces, unless they are nominally flat, with true contact occurring only in isolated patches [38–41]. For surfaces in which microscopic roughness is not significant, previous studies [42,43] reveal that adhesive instabilities are easily triggered in the presence of a cohesive traction law, as to be expected from the JKR model in the limit of infinite radii of curvature. Yet, little has been reported on the jump into and snap out of contact for ideally flat adhesive surfaces, in particular when assuming periodic boundary conditions. In this section, we will be concerned with this question, not only for academic reasons (periodic boundary conditions do not exist in reality), but because this analysis gives clear cues on how to select mesh sizes and how to meaningfully modify CZMs when the mesh size cannot be made arbitrarily small. Towards this end, we use typical energy balance arguments, as originally done by Griffith [44] in the context of cracks and later by Maugis and Barquins [45] in the context of peeling, to describe the force-stress relations in certain

asymptotic limits, while simulations are needed to properly describe those relations near instability points.

Figure 2 shows the stress-displacement relation for a contact described by the two dimensionless numbers $L/\rho = 256$ and $\gamma/(E^*\rho) = 0.15$ along with patterns—as defined by the topography of the elastomer's surface—that arise as stable or metastable solutions. At very large separation, ideally flat surfaces are stable as shown in the inset (a) of Figure 2. When approaching the indenter, the flat configuration becomes suddenly unstable, and a circular bulge, see inset (b), is formed. Upon further reduction of the mean gap, the bulge turns into a line ridge, depicted in inset (c). Next, the ridge develops into a dimple, as shown in inset (d). Finally, the elastomer's surface flattens out again at close approach as revealed in inset (e).

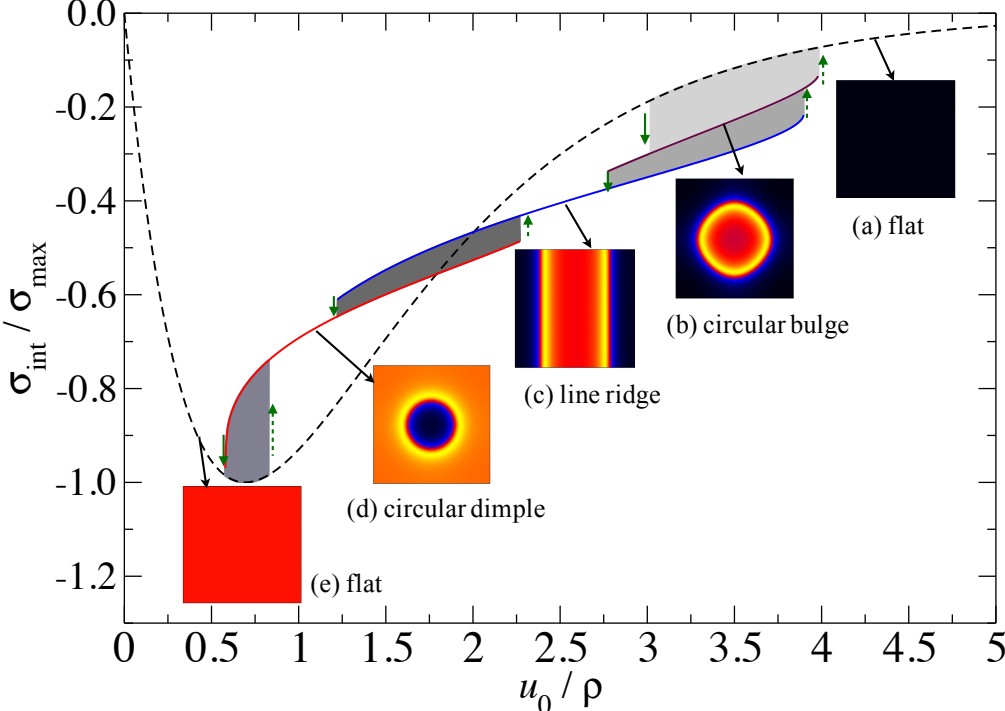

**Figure 2.** Mean stress (normalized to the maximum adhesive strength) as a function of mean displacement (in units of the interaction range) during approach (blue, upper solid curves) and retraction (red, lower solid curves). Four gray regions indicate the energy loss. The square insets show representative local, interfacial gaps on different branches, which increase from red to orange to yellow to blue to black. Solid and dashed red arrows indicate instabilities on approach and retraction, respectively. The dashed line indicates the stress-displacement relation for a flat elastomer.

All transitions shown in Figure 2 are reversible, but discontinuous and thus hysteretic: upon retraction of the elastomer, the patterns reverse, however, at a larger mean gap than during contact formation. The areas between approach and retraction curve in the stress-displacement relation corresponds to the dissipated surface energy. In contrast to ordinary visco-elastic losses, the lost energy depends very weakly on the velocity $\dot{u}_0$ at small $\dot{u}_0$, see also Refs. [41,46,47] linking adhesive losses to (small-scale) instabilities rather than to visco-elastic effects. Since our simulations are thermostatted to a very small temperature, a minor logarithmic rate dependence of the lost energy with tiny prefactors is obtained.

Note that the patterns shown in the insets of Figure 2 occurred at random locations of the simulation cell. However, they were moved to the center of the graphs for aesthetic reasons. Note also that the line ridge is formed parallel to $x$ with the same probability as to $y$, however, it was never observed to form along the diagonal. To represent ridges consistently, we represented them parallel to $y$. Figure 3 depicts the approach-retraction curve for a

system, in which $\gamma$ was kept the same as before, but $L$ was increased to $L = 1024\,\rho$, i.e., to four times the linear dimension of the system represented in Figure 2.

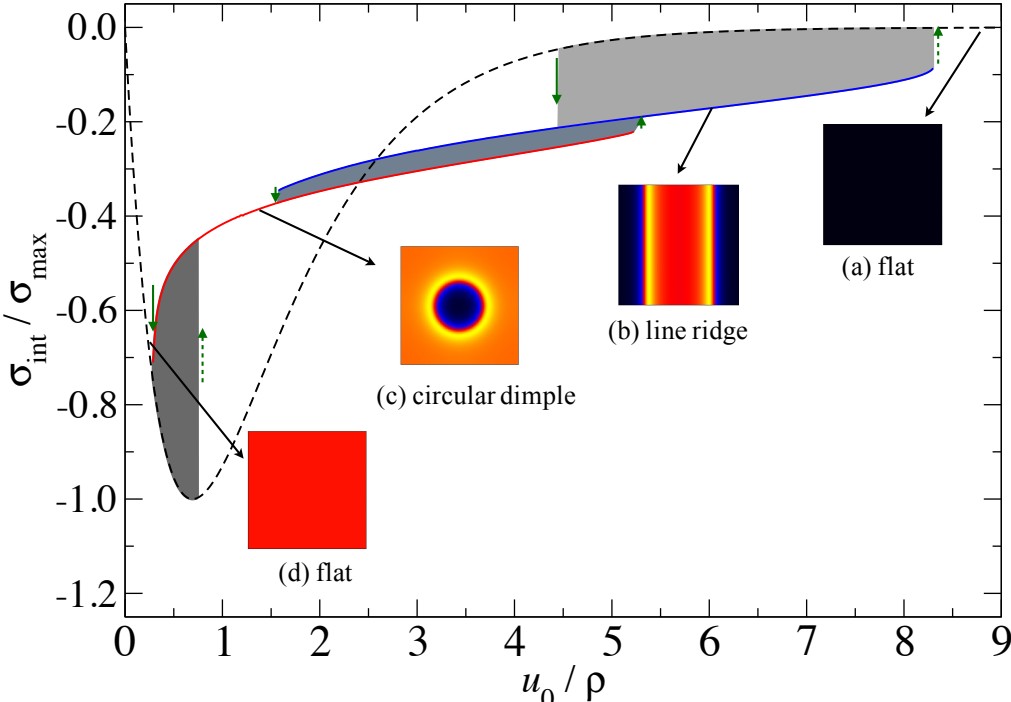

**Figure 3.** Similar to Figure 2, however, for a linear system size of $L = 1024\,\rho$.

While surface patterns and instabilities show similarities for the two different system sizes, notable differences can be observed: in the larger system, the circular bulge has disappeared and instabilities span a broader range in the interfacial displacement than before. In addition, the energy hysteresis per unit area, $\gamma_{\text{hys}} = \oint du_0 \sigma(u_0)$, has grown by a factor close to 4, which means that the total lost energy is still far from a linear scaling with system size for the used appropriate dimensionless numbers describing our system.

In the remaining part of this section, we attempt to rationalize and to quantify the differences for the different system sizes. This is done by two means, first by exploring a harmonic approximation around the stable or metastable, undeformed elastomer. This analysis provides a first guideline for how to set the minimum value for the range of adhesion in a cohesive-zone model-based (peeling) simulation. Second, an energy analysis of the characteristic defect pattern is performed similarly to the traditional Griffith analysis. Ref. [23], however, adopted to periodically repeated domains. As a word of honesty, we must confess that we cannot fully judge to what extent Griffith theory of brittle fracture is simply "reinvented" in some of the following calculations, as we even find text books on that matter somewhat difficult to follow. If it is a reinvention, we hope to have provided an alternative derivation, which is easier to digest than common treatments of that matter, in particular because our treatment is based entirely on the (Fourier) stress-strain relation and does not necessitate any prior knowledge of linear fracture mechanics.

Before proceeding to the theoretical analysis, a few words of clarifications might still be in place. First, it is important to note that controlling the center-of-mass of the layer facing the indenter is impractical experimentally, due to the finite (combined) compliance of the elastomer and the loading apparatus. However, to know the (possible values or range of values of the) traction at a given separation between two (coarse-grained) surface elements, we need to constrain their fully resolved structure at that separation. This why we opted not to run our simulations in a force-controlled fashion. Second, much of what is done in the remaining theoretical section relates to existing literature on configurational instabilities occurring during contact formation [21,22] or delamination [48]. However, we

felt the need to have a coherent description of the various patterns, which is adopted to the surface interactions assumed in this work. At the same time, we explore immediately what can be learned from this analysis for (a) the construction of CZMs and (b) the correction of energy hysteresis from simulations using relatively long-range CZMs to short-ranged CZMs. Last but not least, analyzing how the choice of $\mu_\rho$ affects the result often addresses two questions at the same time. (a) Is $\mu_\rho$ an appropriate choice under the assumption that we aim to model short-range adhesion? (b) How does the range of adhesion affect the system? In this sense, the circular bulge pattern observed in the small simulation cell can be said to have arisen as an artifact or as a consequence of finite-range adhesion.

### 3.1. Harmonic Approximation

At mean gaps, where an undeformed surface is the only stable solution, any deviation of the function $u(x,y)$ from $u(x,y) \equiv u_0$ is counteracted at fixed $u_0$ by a restoring force. For small perturbations, $\gamma(\mathbf{r})$ and therefore also $V_{\text{int}}[u(\mathbf{r})]$ can then be expanded as a second-order Taylor series in the displacement, so that the total excess energy w.r.t. a flat surface reads

$$\Delta V_{\text{tot}} = \frac{A}{2} \sum_{\mathbf{q}, q \neq 0} \left\{ \gamma''(u_0) + \frac{qE^*}{2} \right\} |\tilde{u}(\mathbf{q})|^2 + \mathcal{O}(\delta u^3).  \tag{5}$$

Thus, when $\gamma''(u_0)$ is negative, the harmonic approximation cannot be maintained if there exists a non-zero wave vector whose magnitude is less than the critical wave number

$$q_c(u_0) \equiv -2\gamma''(u_0)/E^*.  \tag{6}$$

In other words, if the linear dimension of a periodically repeated cell exceeds a critical length

$$L_c = 2\pi/q_c,  \tag{7}$$

the surface will deform spontaneously in response to a tiny perturbation of appropriate symmetry.

For fixed system size, two critical separations (may) result. For the used Morse potential, these can be evaluated to

$$u_c = -\rho \ln \left\{ \frac{1}{4} \pm \frac{1}{4} \sqrt{1 - \frac{4\pi\rho^2 E^*}{\gamma L}} \right\}.  \tag{8}$$

Thus, for linear system sizes less than the critical size $L_c = 4\pi\rho^2 E^*/\gamma$, the undeformed surface can remain (meta) stable at any separation and instabilities can be avoided, even if configurations with lower potential energy may exist. Figure 4 confirms that the just-made analytical calculations are consistent with the results of GFMD simulations.

### 3.1.1. Scale-Dependent Cohesive Zone Models

How do the just-obtained results relate to the construction of cohesive-zone models? Assume that a system is discretized to an in-plane linear dimension of $a_0$. If $\rho$ were much less than the critical value below which a periodically repeated cell of length $a_0$ adopts internal defects, then a proper representation of the defect structure (e.g., a peeling front) cannot be represented. Subsequently, the energy required for the peeling process would be much too large. If, however, $\rho$ were much in excess of the critical value, then the adhesion would become long ranged and potentially too long-ranged for a given purpose, e.g., if a system had (microscopic) roughness, or the tape to be peeled were very thin. In that case, the force required to peel the system might be underestimated. This means that the

optimum choice for the mesh size, or, alternatively, the choice of the optimum range of interaction, should satisfy

$$\rho \gtrsim \sqrt{\frac{\gamma \, \Delta a}{4\pi \, E^*}} \tag{9}$$

in the case of Morse potential.

For a general CZM, the just-proposed criterion could also be formulated as

$$\min\{\gamma''(u)\} = -\mu_\rho^2 \cdot \frac{E^*}{\Delta a}, \tag{10}$$

where $\mu_\rho$ should be a constant of order unity. The precise optimum value for $\mu_\rho$ will depend on the specific functional form of the CZM, however, we do not expect a great sensitivity for reasonable choices. In the case of the Morse potential, Equation (10) translates (back) to

$$\rho = \frac{1}{2\,\mu_\rho} \sqrt{\frac{\gamma \Delta a}{E^*}}. \tag{11}$$

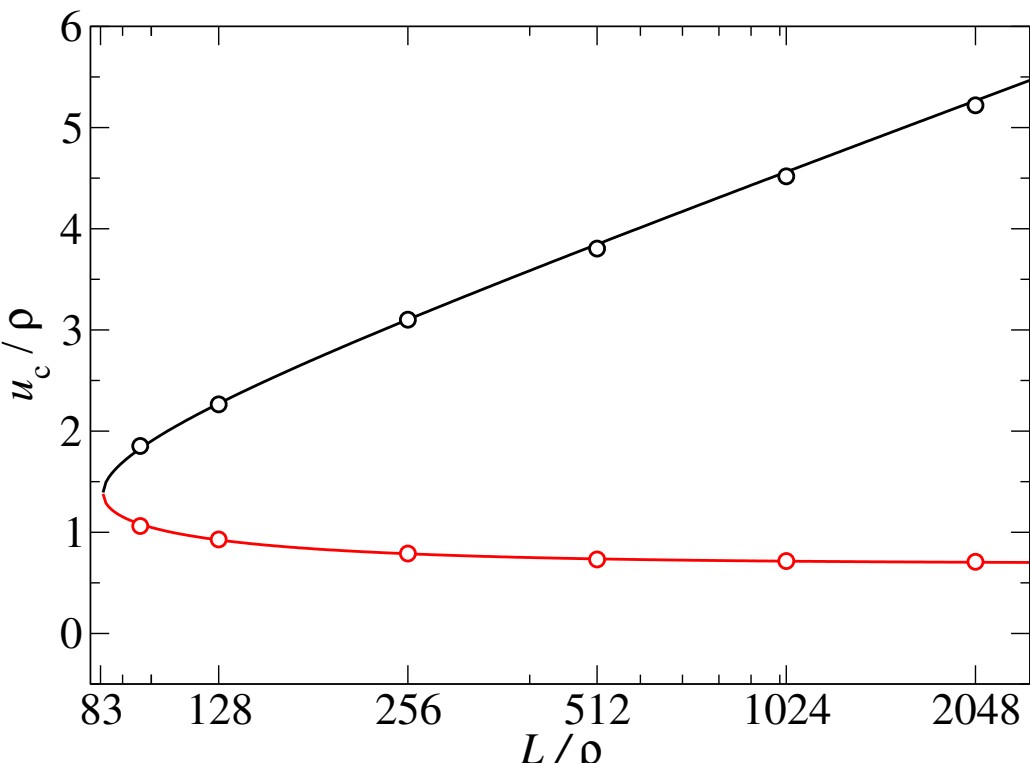

**Figure 4.** Critical separations at which an undeformed, flat surface becomes unstable. The upper (black) and the lower (red) branch relate to approach and retraction, respectively. Circles show GFMD simulation results, while lines reflect Equation (8).

### 3.2. Griffith-Based, Continuum Approach

In this section, we identify some traction-displacement relations for mechanically stable or meta-stable, non-constant displacement fields. Thus, we attempt to minimize the total energy

$$V_{\text{tot}} = V_{\text{ela}} + V_{\text{int}} + V_{\text{ext}} \tag{12}$$

with respect to the displacement field, which contains an "external energy" $V_{\text{ext}}$ in addition to the elastic and interaction energies, which have already been introduced. $V_{\text{ext}}$ is the energy gained in response to an external load, including gravitational loads, i.e.,

$$V_{\text{ext}} = -p_{\text{ext}}\, u_0\, A$$
$$= p\, u_0\, A, \tag{13}$$

where the external pressure $p_{\text{ext}}$ plays the role of a Lagrange parameter, which is adjusted such that the desired mean displacement $u_0$ is an extremum of the total energy. The pressure $p$ exerted from the indenter has the opposite sign of $p_{\text{ext}}$, but is equal in magnitude.

In order to proceed analytically, adhesion is considered to be infinitesimally short-ranged, so that

$$V_{\text{int}} = -\gamma\, A_{\text{c}}, \tag{14}$$

where $A_{\text{c}}$ is the real contact area.

In the following treatment, we will minimize the total energy per area. A lower-case letter $v$ (with varying indices, i.e., ela, ext, int, and tot) will indicate that the pertinent energy is re-expressed as a surface energy density. Moreover, a periodically repeated square domain of length $L$ will be assumed.

Since elasticity is a scale-free theory, in which energy increases quadratically with the displacement, and adhesion is considered to be infinitesimally short ranged, the mean total energy density of a given defect pattern must be of the form

$$v_{\text{tot}} = \frac{E^* u_0^2}{L}\, \hat{v}_{\text{ela}}(\alpha) + p\, u_0 - \gamma\, \hat{a}(\alpha), \tag{15}$$

where $\alpha L$ is the linear dimension of the non-contact with $0 < \alpha \le 1$ so that $\alpha L$ would be, for example, the diameter of a dimple, Moreover, $\hat{v}_{\text{ela}}(\alpha)$ is a dimensionless function of $\alpha$, while $\hat{a}(\alpha)$ denotes the relative contact area, i.e.,

$$\hat{a}(\alpha) = \begin{cases} \pi\, (\bar{\alpha}/2)^2 & \text{(bulge)} \\ \bar{\alpha} & \text{(ridge)} \\ 1 - \pi\, (\alpha/2)^2 & \text{(dimple)} \end{cases}, \tag{16}$$

where $\bar{\alpha} \equiv 1 - \alpha$ is the linear dimension of a contact patch in units of $L$.

The non-trivial part of the calculation is the determination of the function $\hat{v}_{\text{ela}}(\alpha)$. Asymptotic analytical solutions for some defect patterns are derived in the appendix for $\alpha \to 0$ and $\alpha \to 1$. They can also be determined numerically in adhesion-free simulations, as described further below. For the moment, we simply assume the function $\hat{v}_{\text{ela}}(\alpha)$ to exist and to be differentiable.

For any stable solution, both $u_0$ and $\alpha$ must minimize the mean energy density, which is why the partial derivatives of $v_{\text{tot}}$ with respect to these two variables must be equal to zero. Thus,

$$\frac{\hat{v}'_{\text{ela}}(\alpha)}{\hat{a}'(\alpha)} = \frac{\gamma\, L}{E^* u_0^2} \tag{17}$$

$$p = -\frac{2\, E^*\, u_0}{L}\, \hat{v}_{\text{ela}}(\alpha) \tag{18}$$

in mechanical equilibrium. A consequence of Equation (17) is the existence of a maximum (or minimum) displacement $u_0$ if the l.h.s. of Equation (17) has a minimum (or maximum).

Defining $\hat{\Xi}(\dots)$ such that $\alpha = \hat{\Xi}\{\hat{v}'_{\text{ela}}(\alpha)/\hat{a}'(\alpha)\}$ and inserting the resulting value of $\alpha$ into Equation (18) yields

$$\tilde{p} = -2\, \tilde{u}_0^2\, \hat{v}_{\text{ela}}\left\{\hat{\Xi}\left(\tilde{u}_0^{-2}\right)\right\}, \tag{19}$$

after expanding the fraction with $u_0/\gamma$. Here, we used

$$\tilde{u}_0 = \frac{u_0}{\sqrt{\gamma L/E^*}} \tag{20}$$

$$\tilde{p} = \frac{p}{\gamma/u_0}. \tag{21}$$

Thus, for any defect pattern, there is a unique shape of the $p(u_0)$ dependence in the continuum limit, which is obtained by expressing $u_0$ in units of $\sqrt{\gamma L/E^*}$ and $p$ in units of $\gamma/u_0$.

The most important missing ingredient to identify the stress-displacement relation summarized in Equation (19) is the determination of the dimensionless function $\hat{v}_{\rm ela}(\alpha)$. For its numerical determination, we proceeded as follows: For a given defect pattern and a given fixed value of $\alpha$, contact points were defined and constrained to a zero displacement. The energy is minimized with respect to the unconstrained displacement field under a given external pressure $p_{\rm ext}$. In the last step, $V_{\rm ela}$ and $u_0$ are determined from $u(\mathbf{r})$. This was done for different discretizations, which allowed us to perform a Richardson extrapolation of the two observables of interest to the continuum limit for each value of $\alpha$.

In the remaining part of this section, we will present our numerical results on $\hat{v}_{\rm ela}(\alpha)$ and compare them to asymptotic results wherever appropriate, as well as with simulation results that were obtained with finite-range adhesion. Since an accurate determination of $\Xi(\tilde{u}_0^{-2})$ turned out to be very labor intensive, we decided to abstain from this exercise for now.

### 3.2.1. Line Ridge

The line ridge is considered first and with a greater level of detail than the other patterns, since it allows peeling to be studied in the most straightforward fashion. Periodic boundary condition makes the simulation cell have two peeling fronts, which are mirror images of each other.

Two possible asymptotic limits arise, namely a thick ridge with a localized "line crack" as defect pattern for $\alpha \to 0$ and a thin contact ridge for $\alpha$ approaching unity from below as closely as possible. For each limit, it is possible to identify a closed-form analytical expression for $\hat{v}(\alpha)$:

$$\hat{v}_{\rm ela}(\alpha) = \begin{cases} \frac{2}{\pi \alpha^2} & \text{(thick line ridge)} \\ \frac{\pi}{-4\ln(\pi\tilde{\alpha}) + 8c} & \text{(thin line ridge)} \end{cases} \tag{22}$$

with $c = 0.3079(7)$. These two expressions are derived in Appendices A.1 and A.2. Figure 5 reveals that the analytical results for $\hat{v}_{\rm ela}(\alpha)$ are consistent with GFMD data.

As mentioned before, $u_0$ has extrema (and thus end points) when the l.h.s. of Equation (17) has an extremum. Since $\hat{a}'(\alpha) = -1$ for a line ridge, an endpoint of $u_0(\alpha)$ coincides with an extremum in $\hat{v}'_{\rm ela}(\alpha)$. Since $\hat{v}'_{\rm ela}(\alpha)$ is monotonic at small $\alpha$, no unstable point exists in the continuum solution for thick line ridges. Thus, the instabilities in the GFMD simulations toward the formation of dimples can only have arisen due to adhesion having been modeled with a finite range. The power-law relation

$$\alpha = \left(\frac{4E^* u_0^2}{\pi L \gamma}\right)^{1/3}, \tag{23}$$

is easily deduced in the $\alpha \to 0$ thick-ridge limit, which turns out to be quite accurate even up to $\alpha \lesssim 0.7$, as evidenced in Figure 6.

In contrast to the thick-line-ridge limit, the thin-line-ridge asymptote does have a critical value $\alpha_c$, at which $\hat{v}_{\rm ela}(\alpha)$ has zero curvature. It is located at $\alpha_c \approx 0.92(0)$. Although the thick-line-ridge limit appears to match $\alpha_c$ quite well, it fails to produce a truly satisfactory $p(u_0)$ dependence, because the first and the second derivative are not quite as accurate as $\hat{v}_{\rm ela}(\alpha)$ itself.

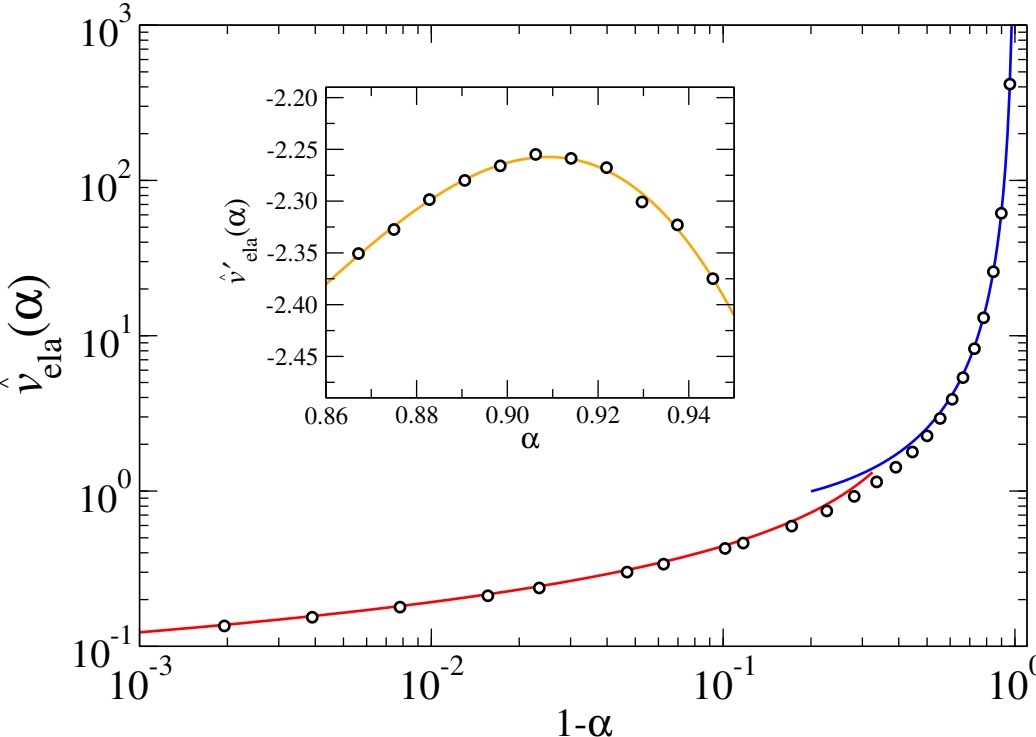

**Figure 5.** Dimensionless elastic energy $\hat{v}_{\text{ela}}(\alpha)$ for a line ridge as a function of $1 - \alpha$. Symbols show GFMD results. The red and blue lines reflect the $\alpha \to 1$ and $\alpha \to 0$ asymptotics respectively. Inset: $\hat{v}'_{\text{ela}}(\alpha)$ in the vicinity of its maximum. The orange line shows a third-order polynomial of $\alpha$.

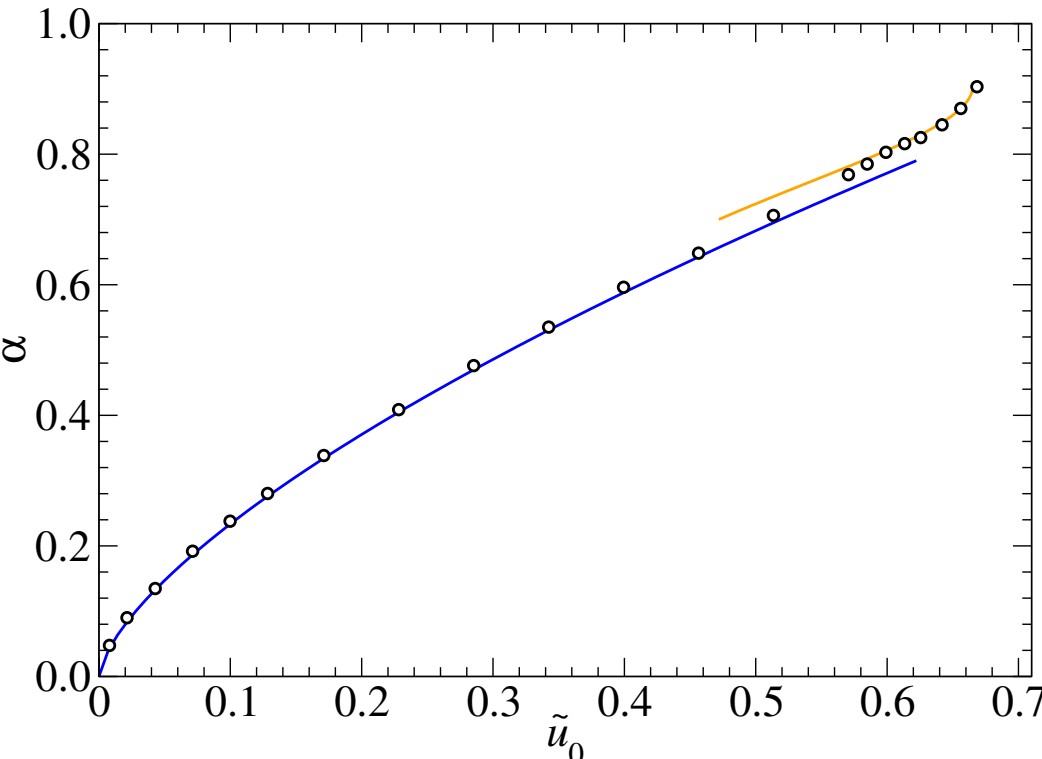

**Figure 6.** Comparison of the $\alpha(\tilde{u}_0)$ dependence obtained with GFMD to the asymptotic thick-ridge (blue line) and critical point (orange line) solutions.

In order to obtain a more precise estimate for the asymptotic thin-ridge behavior before the instability to flattening, GFMD calculations of the reduced elastic energy were refined in

the vicinity of $\alpha_c$. The following results were deduced, which allow that "critical behavior" to be characterized: $\alpha_c = 0.90(9)$, $\hat{v}_c \equiv \hat{v}_{\text{ela}}(\alpha_c) = 0.420(4)$, $\hat{v}'_c \equiv \hat{v}'_{\text{ela}}(\alpha_c) = -2.2(6)$, and $\hat{v}'''_c \equiv \hat{v}'''_{\text{ela}}(\alpha_c) = -1.4(5) \times 10^2$. Thus, near the flattening transition, Equation (17) reads

$$- \hat{v}'_c - \frac{\hat{v}'''_c}{2}(\alpha - \alpha_c)^2 = \frac{1}{\tilde{u}_0^2}, \tag{24}$$

in leading order, which can be easily solved for $\alpha(\tilde{u}_0)$. Just before the flattening instability, a critical separation of $\tilde{u}_c = 1/\sqrt{-\hat{v}'_c} \approx 0.665(6)$ is reached.

The final analytical step is to insert the two analytical $\alpha(u_0)$ dependencies into Equation (18). In the thick-line-ridge limit, this yields

$$\frac{p}{E^*} = -\left(\frac{4\gamma^2}{\pi L E^{*2} u_0}\right)^{1/3}, \tag{25}$$

which reads

$$\tilde{p} = -\sqrt[3]{4/\pi}\,\tilde{u}_0^{2/3} \tag{26}$$

in reduced variables. In the thin-line-limit, we obtain in leading order

$$\tilde{p} = \tilde{p}_c + \tilde{p}_c^{(1/2)}\sqrt{\tilde{u}_c - \tilde{u}_0} \tag{27}$$

with $\tilde{p}_c \approx -0.393(7)$ and $\tilde{p}_c^{(1/2)} = -2/\tilde{u}''_c(\tilde{p}_c) \approx -0.360(0)$.

Figure 7 reveals the correctness of our analysis. The larger system with fixed finite-range adhesion reproduces the continuum solution more closely than the smaller system. This includes a closer approximation of the end-points.

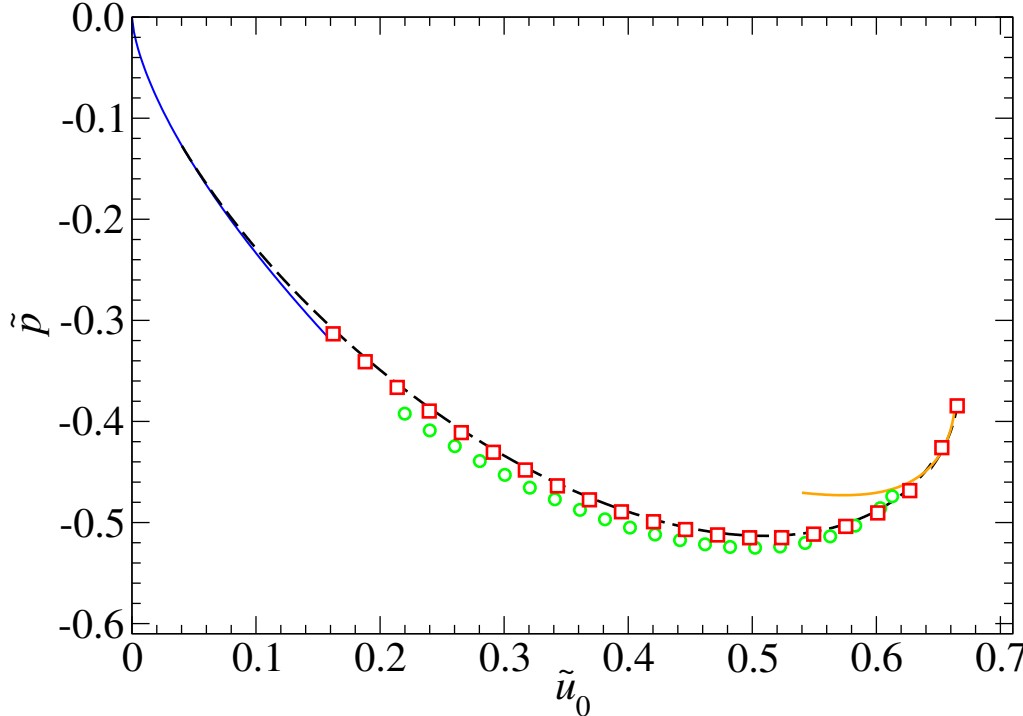

**Figure 7.** Reduced pressure $\tilde{p} \equiv p/(\gamma/u_0)$ as a function of reduced displacement $\tilde{u}_0 \equiv u_0/\sqrt{\gamma L/E^*}$ for different values of $\tilde{\rho} \equiv \rho/\sqrt{\gamma L/E^*}$, i.e., for $\tilde{\rho} = 0.1614$ (green, small circles) and $\tilde{\rho} = 0.0807$ (red, large squares). For these calculations, dimples were suppressed by making the cell in the $y$ direction infinitesimally thin. The full blue and the full orange line represent the thick-line and critical-point asymptotics, respectively, while the dashed black line shows a direct numerical analysis of the GFMD data from Figure 5.

The continuum solution shown in Figure 7 is an overlapping superposition (conglomerate) of three different approaches: On $0 \leq \alpha \leq 0.1$ and on $0.6345 \leq \alpha \leq \alpha_c$ the thick-line-ridge asymptotic solution and the expansion about the flattening point are depicted, respectively. In addition, the GFMD data presented in Figure 5 were processed numerically to yield results on $0.05 < \alpha < 0.663$. It agrees with the two shown approximations within line widths in the shown overlapping domains.

We now turn our attention back to a computational question central to this study. How can we design a CZM such that it reproduces the $\tilde{p}(\tilde{u}_0)$ relation for zero-range adhesion as accurately as possible for a given, fixed number of grid points? In Section 3.1.1, a scaling relation was proposed towards this end, which is tested next. Figure 8 reveals that using $\mu_\rho \gtrsim 0.5$ induces instabilities and thus hysteresis on the $p(u_0)$ curve, which do not exist in the continuum solution and which would disappear if $\rho$ was kept constant but the mesh was refined. For $\mu_\rho \lesssim 0.5$, instabilities disappear but only a relatively small part of the line-ridge solution is stable for the given discretization of $n_x = 16$.

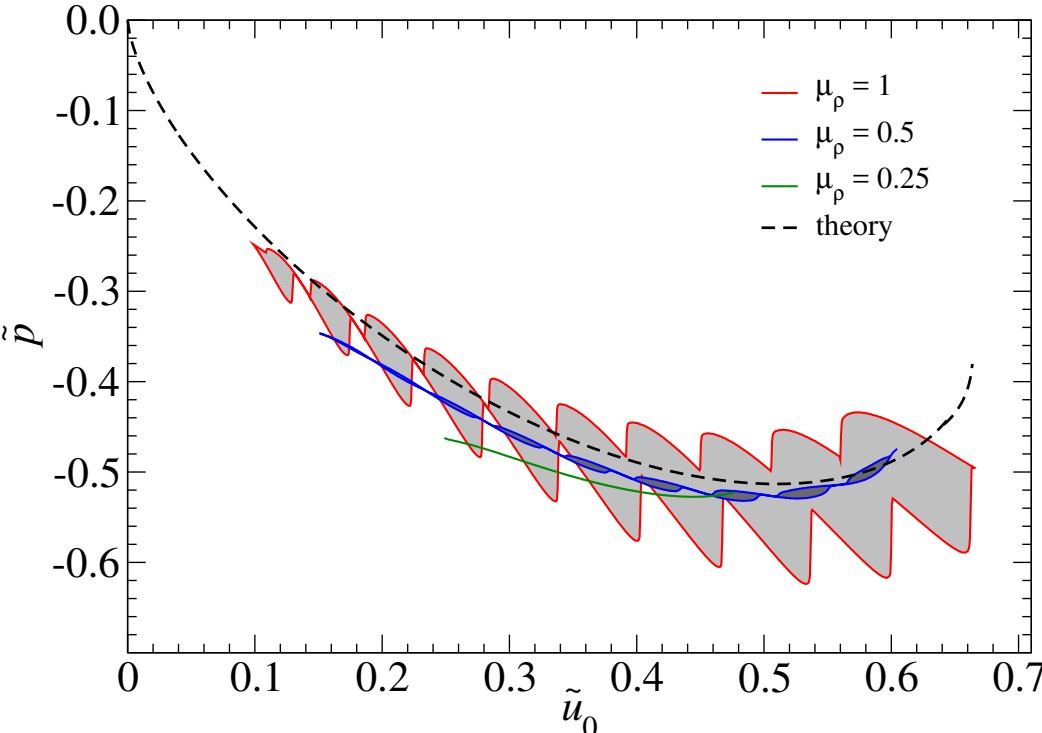

**Figure 8.** Reduced pressure $\tilde{p}$ as a function of reduced displacement $\tilde{u}_0$ for a fixed mesh of $n_x = 16$ grid points in *x*-direction. For these calculations, dimples were suppressed by making the cell in the *y* direction infinitesimally thin. Different scaling parameters $\mu_\rho$ determining the range of interaction were used.

Despite visible discrepancies, the agreement between the exact solution and the one obtained for $\mu_\rho = 0.5$ can be called surprisingly good, because the discretization of the simulation cell into $n_x = 16$ elements disposes only of eight independent, i.e., symmetry-unrelated points to describe contact plus non-contact. They both have fields (stress and derivative of displacement) that cannot be Taylor expanded upon. This makes a total of four fields, which are numerically difficult to integrate, because the simulation cell contains two peeling processes, plus the zones in between the diverging fields. Their combined effect is reflected by merely 16 grid points. Anyone having applied numerical integration schemes to such "poorly behaved" functions will thus certainly appreciate the "performance" of the $n_x = 16$, $\mu_\rho = 0.5$ simulation. Specifically, for $\mu_\rho = 0.5$, the line ridge becomes unstable to flattening at $\tilde{u}_0 \approx 0.15$ for a thick ridge (dimples were suppressed by using $n_y = 1$ for the analysis of ridges) and at $\tilde{u}_0 \approx 0.6$ for a thin ridge. From Figure 6, it

becomes obvious that non-contact is only about 30% of the simulation cell in the first case and contact is only 20% of the simulation cell in the second. At that point, a simulation effectively evaluates an integral over displacement (first case) or stress (second case) field using only two to three integration points. Yet relative errors are relatively small. They decrease quite substantially for all three studied choices for $\mu_\rho$ when the linear mesh size is reduced to half its value. Evidence for this claim is not shown explicitly, because the main problem is the approach *to contact* rather than a proper description of $p(u_0)$ *in contact*, as will be discussed further below.

### 3.2.2. Circular Defect Patterns

Since our main interest is the line ridge, we only sketch results for the two remaining defect patterns. The dimensionless elastic energy for the two circular patterns satisfies

$$\hat{v}_{\text{ela}}(\alpha) = \begin{cases} \frac{8}{\sqrt{3}\pi\,\alpha^3} & \text{dimple, } \alpha \to 0 \\ \sqrt{2}(1-\alpha)^{3/2} & \text{bulge, } \alpha \to 1. \end{cases} \tag{28}$$

Figure 9 shows the numerical results for $\hat{v}_{\text{ela}}(\alpha)$ of the two circular defects, including their asymptotic behavior.

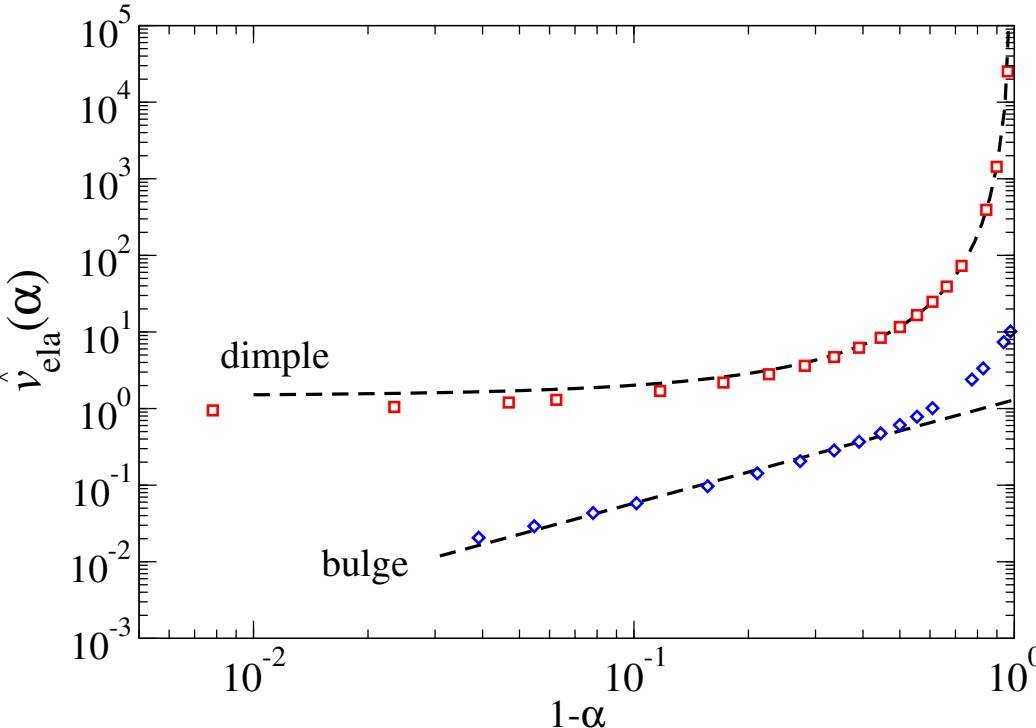

**Figure 9.** Dimensionless elastic energy $\hat{v}_{\text{ela}}(\alpha)$ as a function of the relative, linear contact dimension $\bar{\alpha}$ for the dimple (red squares) and the bulge (blue diamonds).

Proceeding as above, the $\tilde{p}(\tilde{u}_0)$ is obtained as

$$\tilde{p} = -(4\tilde{u}_0/3)^{4/5} \tag{29}$$

for the dimple. Figure 10 reveals that this asymptotic solution is approached as the (dimensionless) range of adhesion is reduced.

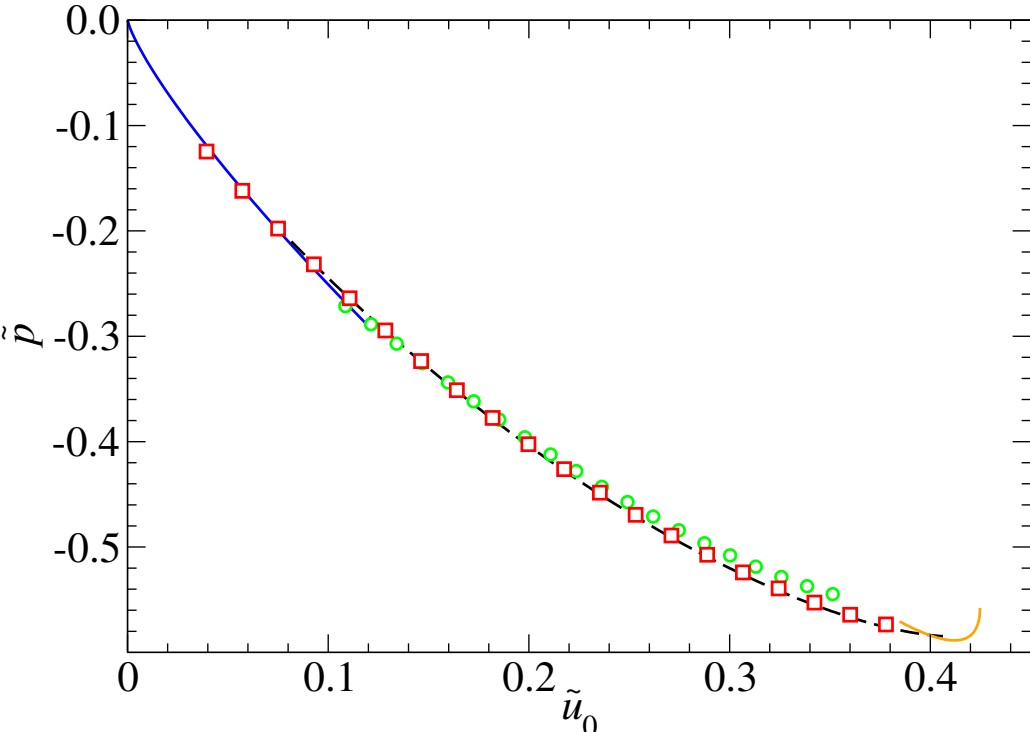

**Figure 10.** Reduced pressure $\tilde{p}$ as a function of reduced displacement $\tilde{u}_0$ for different values of $\tilde{\rho}$. i.e., for $\tilde{\rho} = 0.1614$ (small, green circles) and $\tilde{\rho} = 0.0807$ (large, red squares). The full blue and the full orange lines represent the point-dimple and critical point asymptotics, respectively, while the dashed black line shows a direct numerical analysis of the GFMD dimple data from Figure 9.

No stable solution exists for the bulge in the continuum limit, because an extremum in $v_{\text{tot}}(\alpha)$ is a maximum in $\alpha$. Thus, the bulge in Figure 2 can only have arisen as a consequence of the finite-range of the adhesion. This argument is supported by the bulge's disappearance in Figure 3, in which the (dimensionless) range of adhesion was reduced compared to that used in Figure 2. It is also consistent with the observation that the detachment process of a nominally flat surfaces (which can be roughly mimicked with—or "coarse-grained" to—Morse-like potentials) frequently has one last contact patch in place before the contact breaks.

*3.3. Dissipated Energy*

When two or more stable microstates coexist for a given collective degree of freedom, quasi-discontinuous transitions between them occur when the collective degree is driven externally. This is the mechanism by which multistability leads to instability and ultimately to energy loss, which, as stated in Coulomb's law of friction, predominantly depends on the moved distances and much less on rates or velocities [37,49]. For Coulomb's friction law and related laws to be applicable, the motion has to be slow enough to prevent "basin hopping" between the two stable "macro" states when they are similar in energy, but not so fast that significant heating occurs. In this section, we calculate the energy hysteresis arising from the multistability of non-contact and a line ridge.

In a first approximation, the stress can be approximated with zero as long as the elastomer is flat. The approximation is exact for potentials with a true cut-off, as for example, in the potential introduced later in Equation (31). When the range of adhesion is very small, the elastomer turns directly to a thick line ridge upon approach, which happens at the distance $u_{c,nc}$, where the flat, non-contact solution becomes unstable. It is the larger of the two solutions in Equation (8), that is, the one in which the minus sign is selected in the parenthesis on the r.h.s. of that equation. Upon retraction, the elastomers flattens out

again at the critical distance, $u_{c,lr}$, where the line-ridge solution becomes unstable. Thus, for short-range adhesion

$$\oint du_0 \, \sigma(u_0) \quad \approx \quad \int_{u_{c,nc}}^{u_{c,lr}} du_0 \, \sigma_{lr}(u_0) \tag{30a}$$

$$\approx \quad \frac{3}{2} \left( \frac{4\gamma^2 E^*}{\pi L} \right)^{1/3} u_0^{2/3} \Bigg|_{u_0 = u_{c,nc}}^{u_{c,r}} \tag{30b}$$

$$\approx \quad \frac{3}{2} \alpha_c \, \gamma - \frac{3}{2} \left( \frac{4\gamma^2 E^*}{\pi L} \right)^{1/3} u_{c,nc}^{2/3} \tag{30c}$$

$$(\text{for Morse}) \quad \approx \quad \frac{3\gamma}{2} \left\{ \alpha_c - \left( \frac{2\tilde{\rho}}{\sqrt{\pi}} \ln \frac{2}{\pi\tilde{\rho}^2} \right)^{2/3} \right\}. \tag{30d}$$

is obtained in a cycle going from non-contact to line ridge and back to non-contact.

In a more refined calculation, the "integration constant" $3\alpha_c/2$ can be replaced with a more precise value for the lost energy in the continuum limit. The latter is best obtained by integrating (numerically) the $p(u_0)$ curve that is reconstructed from the reference line shown in Figure 7. Moreover, a correction of $v_{int}(u_{c,nc}) - v_{int}(u_{c,lr})$ due to the gained energy while approaching the counterface in non-contact must be subtracted from the dissipated energy to yield accurate estimates.

The second term on the r.h.s. of Equation (30c) is the main correction to the dissipated energy that arises by replacing a zero-range with a finite-range adhesion. Unfortunately, convergence of the computed dissipated energy is rather slow. For CZMs with a true cutoff $g_c$ linear in $\rho$, the error disappears with $\rho^{2/3}$ and thus with $\Delta a^{1/3}$. For the Morse potential, this scaling is further impeded by corrections logarithmic in $\rho$. GFMD data confirm these conclusions in Figure 11.

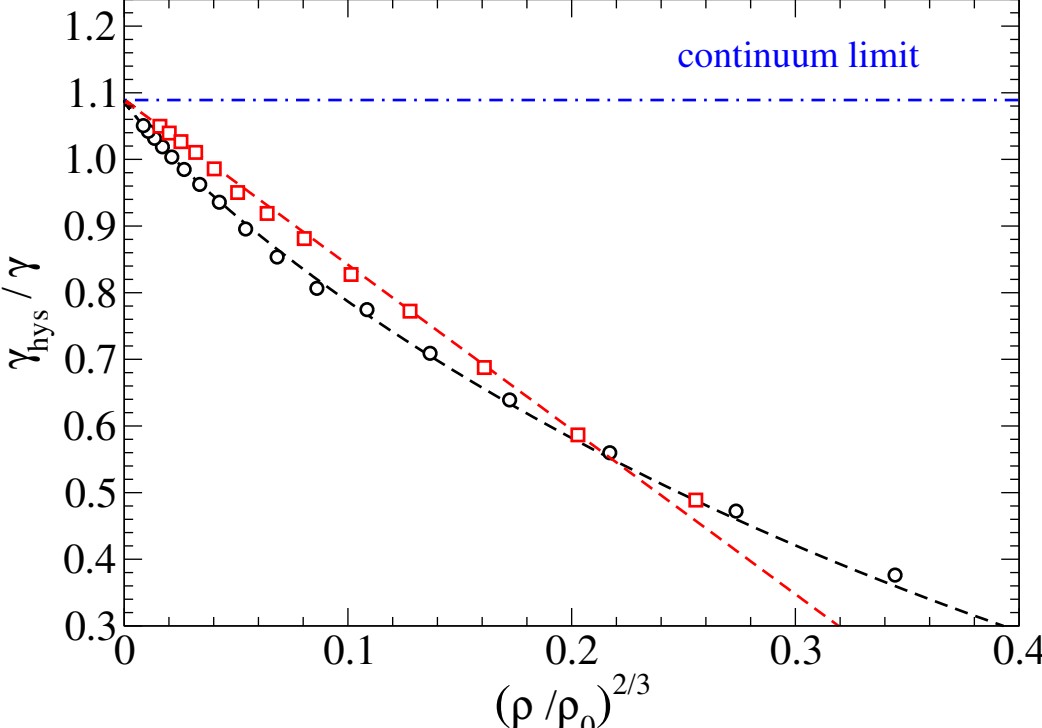

**Figure 11.** Normalized dissipated energy $\gamma_h/\gamma$ as a function of the dimensionless range of interaction $\rho/\rho_0$ with $\rho_0 = \sqrt{\gamma L/(4\pi E^*)}$. The red and black dashed lines show the theoretical line derived from Equation (30c)—plus the contribution $\gamma(u_{c,nc}) - \gamma(u_{c,r})$ for the Morse potential. Circles and squares indicate GFMD results for Morse potential and cosine potential, respectively. The blue line gives the asymptotic value derived from the analysis of the dashed line in Figure 8.

Since optimizing prefactors is particularly important when convergence is slow, it may be desirable to use other CZMs than the one based on the Morse potential. For a CZM used to study not only (quasi-) statics, as in this work, but true dynamics, an additional requirement would be that the stress is a continuous function of the interfacial separation. This is because (strongly) discontinuous forces or stresses, as they occur in many CZMs at small $g_c$ [4,5,13,14,17], violate energy conservation even for a symplectic integration scheme [50]. This in turn is likely to lead to undesirable dynamical artifacts, e.g., when modeling reciprocating motion. A simple CZM avoiding discontinuous forces is:

$$\gamma_{\cos}(g) = -\gamma \times \begin{cases} 0 & \text{for } g_c < g \\ \frac{1}{2}\{1 + \cos(\pi g/g_c)\} & \text{for } 0 < g < g_c \\ \{1 - (\pi g/g_c)^2/2\} & \text{for } g < 0. \end{cases} \tag{31}$$

Figure 11 reveals that the alternative CZM converges to its asymptotic value more quickly than the Morse potential. Even more importantly, extrapolation to short-range adhesion can be achieved already at relatively large interaction ranges. This is mainly because the alternative CZM lacks the corrections in the second term on the r.h.s. of Equation (30d) that are logarithmic in $\tilde{\rho}$.

## 4. Application to Uneven Surfaces

In this section, we explore to what extent the insights gained for adhesive hysteresis and the modeling of adhesive hysteresis in ideally flat contacts extend to uneven surfaces. To this end, we simulate adhesive contacts with Hertzian and nominally plane indenters. While our initial motivation for these simulations was to explore how the continuum limit can be approached in the most effective way, it is also possible to look at these calculations as if the used CZMs had arisen from integrating out the effect of small-wave-length surface undulations, i.e., from wave lengths much less than either the contact radius in a Hertzian contact geometry or less than the short wave length cutoff in the simulation on nominally flat surfaces.

### 4.1. Application to Hertzian Contacts

We consider a Hertzian contact with radius of curvature $R_c$ and contact modulus $E^*$, which define the units for length and pressure, respectively. The interfacial energy density, as defined in Equation (3), is assigned the value of $\gamma = 0.59 \times 10^{-3} E^* R_c$. This choice makes the critical contact radius at the pull-off instability be roughly 10% of the radius of curvature, which was also used as the linear size of the periodically repeated simulation cell. This way, the contact radius is small compared to half a cell dimension, so that the periodic boundary conditions have a marginal effect on the contact, while, at the same time, a Fourier-based code remains efficient. Using the definition of the Tabor parameter $\mu_T$ as in Equation (8) of Ref. [51], the relation between $\mu_T$ and $\mu_\rho$ is

$$\mu_T = 2\,\mu_\rho \left(\frac{\gamma}{R_c E^*}\right)^{1/6} \sqrt{n_x}, \tag{32}$$

which turns out to be $\mu_T \approx 0.579 \cdot \mu_\rho \sqrt{n_x}$ for the used parameters. This relation is useful to know for our later analysis. Moreover, we define the displacement such that a flat elastomer, which touches the indenter in its most extreme point is assigned a (mean) displacement of $u_0 = 0$.

Figure 12 compares the displacement-driven force-distance dependence in approach and retraction for different choices of $\mu_\rho$ and varying mesh sizes $\Delta a = L/n_x$. Qualitatively different types of behaviors are produced by using different numerical values for $\mu_\rho$ in Equation (11): (a) If $\mu_\rho$ is small, i.e., less than 0.5, the only observed instabilities are collective jump-into and jump-out-of contact. In this case, the hysteresis compared to the exact solution is strongly underestimated at a coarse discretization, however, the true

hysteresis is approached when increasing $n_x$. (b) As $\mu_\rho$ increases to values around unity, small-scale instabilities occur, which are related to individual rings of (coarse-grained) atoms. The correct hysteresis is still approached, because instabilities on the compressive branch become smaller with increasing $n_x$. (c) As $\mu_\rho$ increases to 1.5, the computed dissipated energy in a compression/decompression cycle starts to depend quite sensitively on how far the system is compressed, e.g., if it is compressed to a zero displacement or to a zero load. For $\mu_\rho = 1.5$, it is not clear if convergence to the continuum limit can be reached. (d) For even larger $\mu_\rho$, small-scale instabilities dominate and both pull-off force, as well as dissipated energy no longer converging to the correct values as the mesh size is decreased.

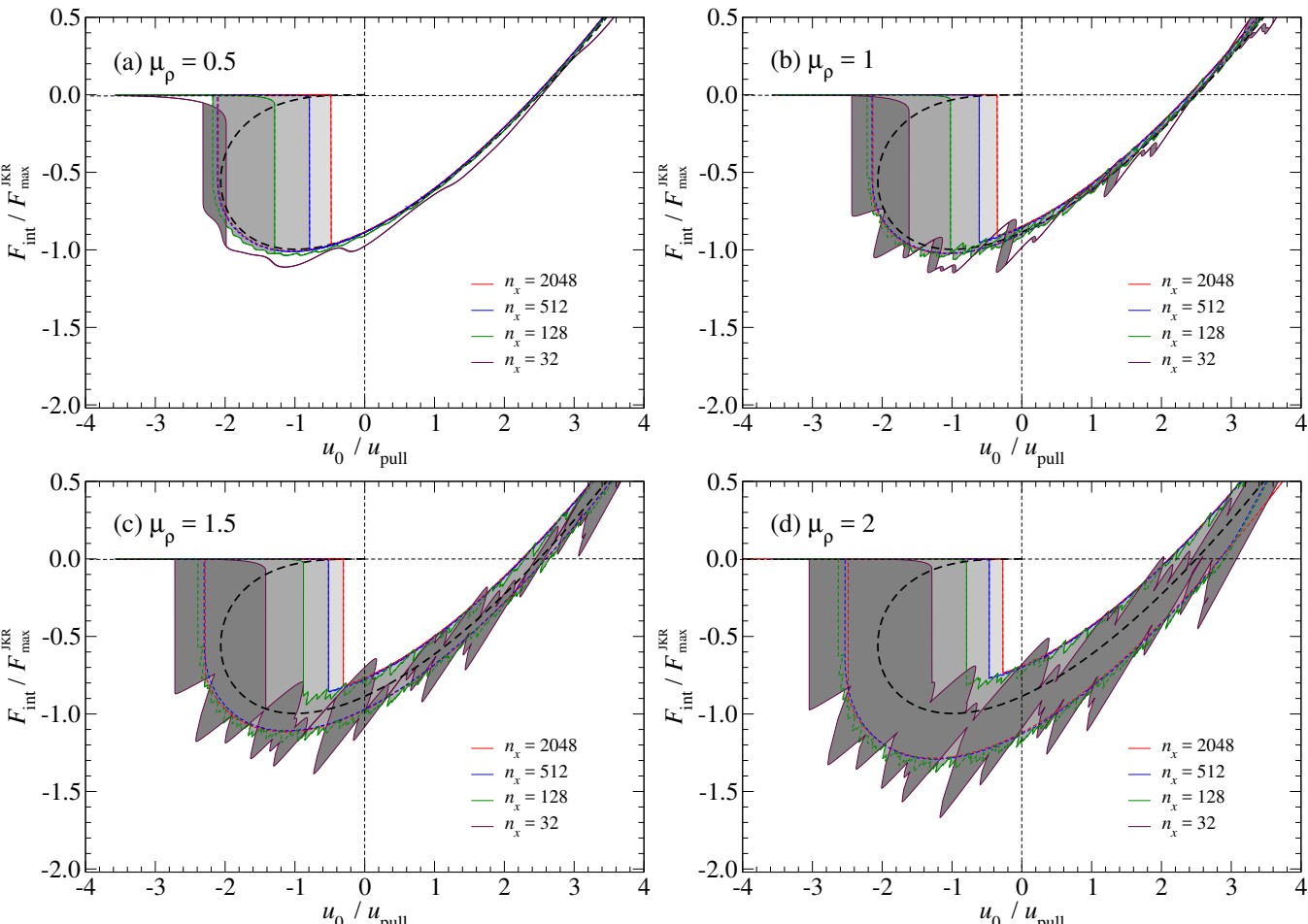

**Figure 12.** Typical traction-separation curves for adhesive Hertzian indenters with different discretization $n_x$ and different scaling factors $\mu_\rho$ determining the range of interaction through Equation (11). (**a**) $\mu_\rho = 0.5$, (**b**) $\mu_\rho = 1$, (**c**) $\mu_\rho = 1.5$, and (**d**) $\mu_\rho = 2$. The used cell dimension $L$ was identical to the radius curvature $R_c$.

The results presented in Figure 12 resemble, to a significant degree, simulations of contacts involving a curved ridge to which a single-sinusoidal undulation is added, see Figures 5–7 in Ref. [46]. In those figures, the force-displacement relation also transits from subtle perturbations of a smooth JKR dependence to violent zig-zag motion. Differences are that our undulations arise from discreteness effects while those in Ref. [46] are due to continuous undulations. Moreover, spacings between discontinuities are irregular in our case but regular in Ref. [46], as our system is two-dimensional, in which case, rings of discretization points have irregular spacings, which, moreover, become smaller and smaller the greater the distance from the symmetry axis.

An interesting feature revealed in Figure 12 is that the JKR separation curve can be approximated quite well with Tabor parameters as small as $\mu_T \approx 1.6$, as evidenced by the $n_x = 32$ curve in Figure 12a. In fact, for $\mu_T = 4$, both the dependence of contact area

and of displacement on load are almost indistinguishable from the exact JKR solution [8] when using large $n_x$ but fixed $\mu_T$. However, the approach curve is still relatively crude even when the Tabor parameter is as large as $\mu_T \approx 10$, i.e., for the ($n_x = 256$, $\mu_\rho = 1$) data set shown in Figure 12b. This confirms a previous analysis by Ciavarella et al. [24], who found a similar dependence of the hysteresis loop on the Tabor parameter as we do, e.g., approximately 50% "error" at $\mu_T = 5$ as can be deduced from their Figure 7.

To further discuss the ramifications of Figure 12, it is useful to know that the critical contact radius in a load-driven separation is $a_c \approx 0.1278 R_c$ for the parameters used, which reduces to roughly half that value of $a_c \approx 0.06315 R_c$ in a displacement-driven separation. Thus, to obtain estimates within approximately 20% accuracy for pull-off stress and dissipated energy density, the length into which the elastomer is discretized should not exceed $a_c/10$ for the given value of $\gamma/E^* R_c = 0.5859 \times 10^{-3}$. This is a finer discretization than for non-adhesive contacts, where we observe an error of order 10% in the normal displacement for a linear mesh size of $\Delta a = a_c/5$.

We next quantify the effect of mesh size on the pull-off force $F_{\text{pull}}$ and on the energy, $V_{\text{hys}} = \oint du_0 F(u_0)$, dissipated in a single c/d cycle. Due to the presence of micro-scale instabilities during contact, $V_{\text{hys}}$ does not have a unique value but depends on the maximum displacement during the compression cycle. We chose it to be the displacement at which the normal load, needed to keep the elastomer at a fixed center-mass, disappeared. In other words, $V_{\text{hys}}$ corresponds to be the gray-shaded areas in Figure 12 below the $x$-axis times the maximum JKR tensile force to which $F_{\text{pull}}$ was normalized. Results for $F_{\text{pull}}$ and $V_{\text{hys}}$ are shown in Figure 13.

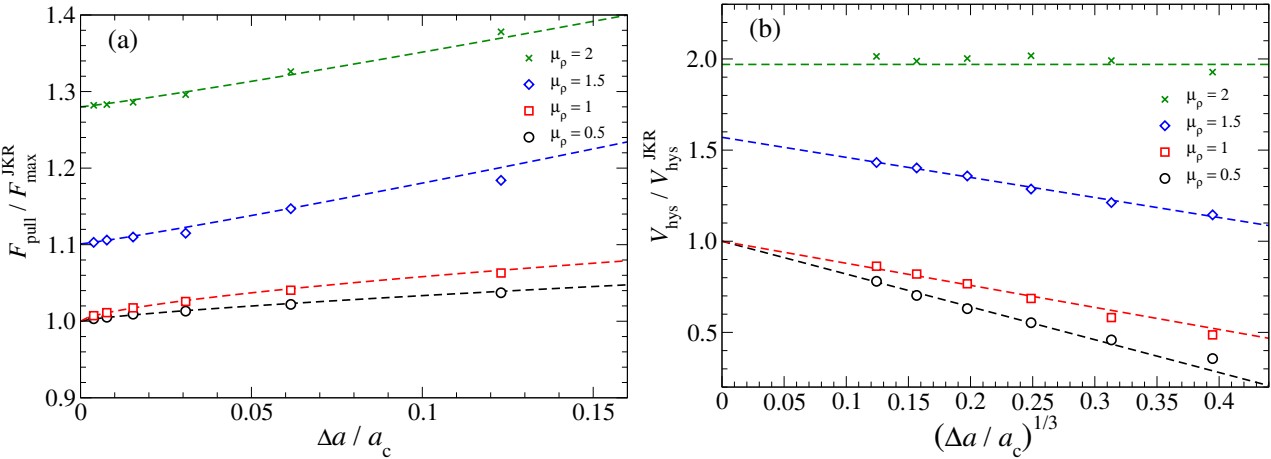

**Figure 13.** (**a**) Maximum tensile force $F_{\text{pull}}$ and (**b**) dissipated energy $V_{\text{hys}}$ as a function of the mesh size $\Delta a$. $F_{\text{pull}}$ and $V_{\text{hys}}$ are normalized to the values deduced from the JKR solution, while $\Delta a$ is normalized to the critical JKR contact radius in a displacement driven separation.

The adhesive Hertzian indenter shows similar behavior as the flat-on-flat geometry in the following ways: the dissipated energy converges noticeably slower to its asymptotic value than the pull-off force. The scaling factor $\mu_\rho$ has to be sufficiently small in order for convergence to the correct values to be reached. For large $\mu_\rho$, results are quite insensitive to the mesh size $\Delta a$. For Hertzian contact geometries, we did not repeat the simulation by replacing the default Morse expression for the surface energy, $\gamma(g)$, with $\gamma_{\text{cos}}(g)$. However, we are certain that convergence to the continuum limit can been reached more quickly with this alternative form.

We note that it can be difficult to judge if the use of a given $\mu_\rho > 0.5$ is not too "aggressive". The data for $\mu_\rho = 1.5$ show the correct trends in the sense that the $F_{\text{pull}}$ decreases and $V_{\text{hys}}$ increases with decreasing mesh size $\Delta a$. However, neither asymptotic quantity converges to its exact value, which can be deduced from the JKR solution, even if errors are not large. Thus, to be sure about the asymptotic values, either $\mu_\rho$ has to be

chosen sufficiently small from the beginning, or it has to be set to two different numbers, which then yield identical asymptotics.

### 4.2. Application to Nominally Plane Contacts

Our final simulations are concerned with nominally flat but randomly rough surfaces. Their topography has been described numerous times in the literature [40,52–55], which is why we abstain from explaining the terms needed to define a height spectrum. It should suffice to state that we used a smooth roll-off with $L/\lambda_r = 4$ and $\lambda_r/\lambda_s = 16$, where $\lambda_r$ and $\lambda_s$ are the roll-off wavelength and short-wavelength cutoff, respectively. The height profile was rescaled to make the root-mean-square height gradient unity. The surface energy was set to $\gamma = 0.3578\, E^* R_c\, \bar{g}^3$, which is close to being large enough to make the surface locally sticky for a Tabor parameter of $\mu_T = 1$ according to the criterion proposed by Pastewka and Robbins (PR) [39] and assuming the PR criterion was worked out correctly in Ref. [51]. The PR criterion states that self-affine surfaces start to be sticky when the surface energy exceeds $\gamma_{PR} = E^* R_c \bar{g}^3 \mu_T^{3/7}$. Here, $\bar{g}$ is the root-mean-square height gradient, and $R_c = 2.12\,\lambda_s$ is a measure for the local surface curvature, as defined in Eq. (6) of Ref. [51]. Additional surface specifications are a root-mean-square height of $5.2\,\lambda_s$ and a maximum height difference of $\Delta h_{max} = 21.6\,\lambda_s$ between the highest and lowest point in the simulation cell.

Compression/decompression cycles were run for different scaling factors $\mu_\rho$. Figure 14 reveals similar trends for the randomly rough surface as for the Hertzian indenter: When $\mu_\rho$ is large, the hysteresis is overestimated. In this case, the lost energy in a c/d cycle is insensitive to the mesh size $\Delta a$ and the continuum limit cannot be reached by decreasing $\Delta a$. The continuum limit can only be approached when $\mu_\rho$ is below a critical value. In that case, the hysteresis is initially rather small and approaches the short-range-adhesion limit quite slowly, while the pull-off force converges much more quickly.

Two additional interesting observations can be made in Figure 14 pertaining to technical issues. First, while the $\mu_\rho = 2$ set of simulations presumably does not reach the exact continuum limit for $\varepsilon_c \equiv \Delta a/\lambda_s \to 0$, it produces quite similar results as the $\mu_\rho = 1$ systems at their finest considered discretization of $\varepsilon_c = 1/16$. Thus, it seems as though the optimum choice for $\mu_\rho$ for the specific problem is just a nudge less than two. Second, the largest considered range of interaction, i.e., $\rho = 0.235\,h_{rms}$ (for $\mu_\rho = 0.5$ and $\lambda_s/\Delta a = 4$), is relatively small compared to the root-mean-square height and only 4% of $\Delta h_{max}$. Yet, the energy hysteresis is rather small. Thus, using interaction ranges that are small compared to $\Delta h_{max}$ or even to $h_{rms}$ is no generally valid criterion for the interaction range to be in the short-range limit. A relative close approach to short-range-adhesion limit requires the range of adhesion to be reduced to $\rho \approx 0.03\,h_{rms}$ for our system. The combination of $\mu_\rho = 1$ and $\lambda_s \gtrsim 32\,\Delta a$ appears to be in that limit.

While the just-made observations are merely methodological in nature, two additional observations can be made with a potential value for our understanding of adhesive hysteresis. First, signs for adhesion on approach become smaller as the range of adhesion is reduced, while the opposite is the case on retraction. In fact, for the simulation in which $\mu_\rho = 1$ and $\lambda_s = 64\,\Delta a$, see Figure 14b, signs of adhesive traction have essentially disappeared on approach, but are clearly visible on retraction. For this trend to be fully revealed, the adhesion must be made very short ranged, i.e., the local Tabor parameter must be equal to or exceed a value of $\mu_T \approx 4$ in our setup. Second, contact hysteresis appears to correlate quite well with the PR criterion. The longest interaction range leads to $\gamma = 0.672\,\gamma_{PR}$. This is just a little below the surface energy at which the PR criterion claims stickiness, and the hysteresis is very small indeed. As the mesh size is decreased, hysteresis becomes noticeable. Thus, the transition between sticky and non-sticky occurs coincides with a surface energy that is close to $\gamma_{PR}$.

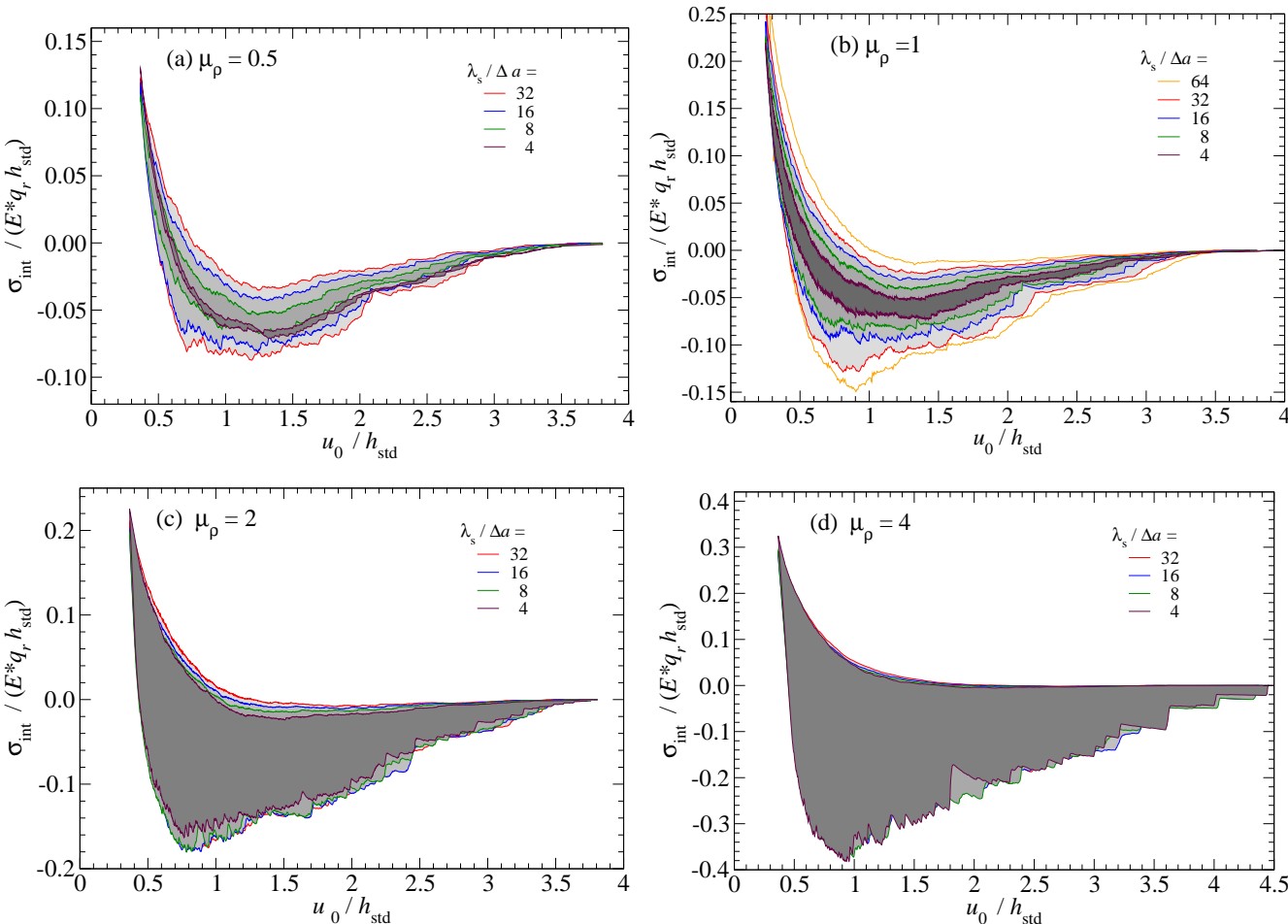

**Figure 14.** Typical traction-separation curves for a randomly rough indenter using different discretization $n_x$ and different scaling factors $\mu_\rho$ of rough indenter determining the range of interaction through Equation (11). (**a**) $\mu_\rho = 0.5$, (**b**) $\mu_\rho = 1$, (**c**) $\mu_\rho = 2$ and (**d**) $\mu_\rho = 4$. For the given system, the range of adhesion is $\rho \approx 0.235\, h_{\text{rms}} \sqrt{\Delta a / \lambda_s} / \mu_\rho$ and the local Tabor parameter $\mu_T = 0.672\, \mu_\rho \sqrt{\lambda_s / \Delta a}$, where $n_x$ is the number of grid points in each spatial direction. Local discontinuities in the stress-displacement curves are due to small-scale instabilities, such as the quasi-discontinuous contact loss or contact formation of contact at maxima or saddle points in the (relative) height topography.

Of course, the just-reported correlation between contact hysteresis and the prediction by the PR criterion may merely be a coincidence. However, previous simulations, in which $\gamma$ was varied rather than the range of adhesion also indicate a transition from sticky to non-sticky at the point, where the reduced surface energy was united [51]. In that work, the transition was observed in the contact area relaxation function, which went from underdamped at $\gamma / \gamma_{\text{PR}} < 1$ to critically damped as $\gamma / \gamma_{\text{PR}}$ approached unity. Yet, system sizes in this and in former work may be too small to draw valid conclusions in favor of the PR criterion. In contrast, the flawed contact-geometries in so-called bearing-area models (BAMs) would render BAM-model arguments against it questionable. Since the PR criterion is a controversially discussed topic in its own right, we postpone its in-depth discussion to the future, in particular in light of the convincing arguments, simulations, and experiments against its validity [12,25,26].

For completeness' sake, we do not want to leave it unmentioned that previous studies found similar traction-displacement dependencies for nominally flat surfaces. This includes, but is supposedly not limited to, early pioneering simulations by Carbone et al. [56] and recent work by Radhakrishnan and Akarapu [57], both finding significant contact hysteresis. However, these two studies considered one-dimensional contact lines, which have a relatively large emphasis on long wavelength undulations, making hysteresis more

easily observable than for two-dimensional interfaces. In contrast, the simulations by Joe et al. [11,12] were concerned with the more-difficult-to-treat two-dimensional interfaces, but this time for adhesion that was long-ranged on the scales of the short-wavelength-cutoff, which again is simpler to treat than short-range adhesion.

In the context of randomly rough surfaces, we do not want to leave it unmentioned either that the maximum system size used in the recent finite element (FE) study [57] was 16,384 grid points, while our largest simulations, which were run (for six weeks) on a 5-year-old commodity computer, contained 8192 × 8192 grid points, which is 4096 times the size of what appears to be manageable using FE methods. These numbers might indicate that mass-weighting GFMD [35] is an efficient method for adhesive boundary-value problems.

Finally, we note that our results for long-range adhesion resemble those by Joe et al. [11,12], who identified an elegant and yet elaborate way to integrate out the effect of short-wavelength roughness. We find that the results that we obtain can be fit reasonably well with the superposition of two exponential functions, in which the prefactor is strongly reduced w.r.t. that of the microscopic interaction law and the length scales appearing in the exponentials of the microscopic law are substantially increased. These results, which are presented in Figure 15, are not very sensitive to the precise choice of the local interactions, i.e., similar coarse-grained force-displacement curves are obtained for a microscopic Morse potential and the easier-to-use alternative CZM defined in Equation (31), as long as their Tabor parameters are similar.

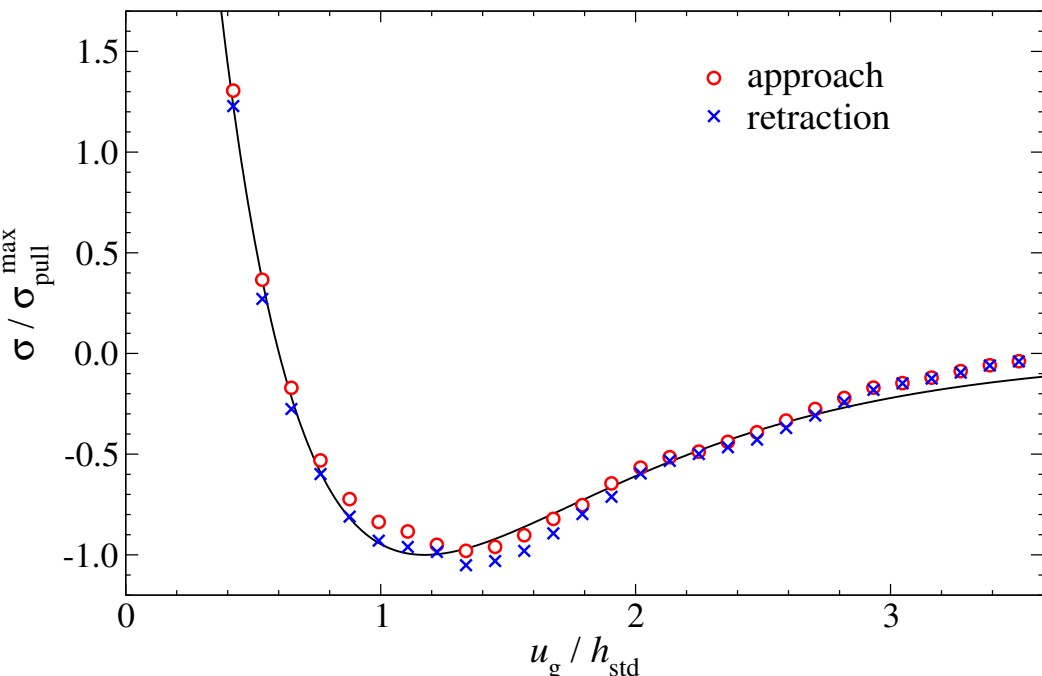

**Figure 15.** Measured cohesive zone-law resulting from a simulation between, for approach (red circles) and retraction (blue crosses). The full line is a fit to the mean value of the compression and decompression curve, which assumes two exponential functions. Compressive stress is expressed in units of the (fitted) maximum tensile stress, and the mean gap $u_g$ is stated in units of the height standard deviation.

## 5. Conclusions and Outlook

The three main aims of this paper were (i) to provide a comprehensible theoretical framework describing the formation and failure (brittle fracture) of an adhesive, periodically repeated interface under constant normal stress and the subsequent energy hysteresis, (ii) to deduce generally applicable rules for the construction of cohesive zone models from the theoretical framework, and (iii) to apply the schemes obtained for the contact between two ideally flat surfaces to uneven surfaces. This last point includes rules for

how to extrapolate dissipation computed with relatively long-ranged adhesion to shorter-ranged adhesion.

A particular focus of our work was the much overlooked approach to contact and the question at what separation an initially flat elastomer approaching a substrate with short but finite-range adhesion becomes unstable to the formation of surface undulations. This happens when the negative curvature of a cohesive-zone model (CZM) exceeds $qE^*/2$, where $q$ is the wavenumber associated with a surface undulation. The ramification for the numerical modeling is that a mesh size should not exceed the scale within which an elastomer would want to ripple during the approach to contact, which leads to the condition

$$\max\{-\gamma''(g)\} \lesssim \frac{E^*}{\Delta a}, \tag{33}$$

$\Delta a$ being the mesh size of an element into which the surface is discretized. This inequality can be used to either define the mesh size or to (re)define the CZM. In this work, we used it to set the range of interaction $\rho$ in a CZM whose functional form was that of the Morse potential, which yields the proportionality $\rho \propto \sqrt{\gamma\Delta a/E^*}$. Using a proportionality factor of $\mu_\rho = 0.5$, see also Equation (11), no undesired instabilities show on the approach curve, while they do occur for $\mu_\rho = 1$.

The usual procedure when modeling adhesive contacts is to ask the question at what tensile stress a mesh element is going to detach [5,13,58]. The common argument is that it does so when the energy released during the detachment process exceeds the work of adhesion, which in turn leads to the condition $\sigma_{\text{tens}}^{\max} \approx \sqrt{2E^*\gamma/\Delta a}$, which—when applied to a continuous, twice differentiable CZM—can be readily translated to Equation (33). Table 1 gives a summary of choices made by different authors, however, translated to the proportionality factor $\mu_\rho$ used for the Morse potential.

**Table 1.** Values for $\mu_\rho$ implicitly used in different cohesive zone models.

| Model | Year | $\mu_\rho$ |
|:---:|:---:|:---:|
| Dugdale [4] | (1960) | 0.798 |
| Hillerborg [15] | (1976) | 0.5 |
| Irwin [16] | (1997) | 0.886 |
| Falk et al. [17] | (2001) | 0.532 |
| Hui et al. [18] | (2003) | 1.09 |
| Popov et al. [19,20] | (2015) | 0.729 |

A compromise needs to be made when choosing the prefactor $\mu_\rho$. For the approach curve, the proportionality factor is chosen at best as small as possible. However, when it is made too small, artificial instabilities and thus energy hysteresis ensue that are not present on the continuum solution. Unfortunately, if the proportionality factor is above a critical value, the continuum solution cannot be reached even for $\Delta a \to 0$. Thus, a relatively safe choice should be to set the prefactor, such that a flat-on-flat geometry reveals no undesired instabilities. It appears as if excellent choices have been made in the literature so that the range of interaction is made small enough to lead to the (almost) best possible convergence, while being large enough to converge to the correct value.

The trouble of Equation (33), as it comes to modeling adhesion in the zero-range or continuum limit, is that the range of adhesion can only be chosen as $\rho \propto \sqrt{\Delta a}$. This poor scaling is particularly problematic for the determination of adhesive hysteresis, because the lost energy density $\gamma_{\text{hys}}$ scales only with $\rho^{2/3}$, so that $\gamma_{\text{hys}}$ has corrections that cannot disappear more quickly than with $O(\sqrt[3]{\Delta a})$, which for a two-dimensional surface implies an $O(\sqrt[6]{N})$ converge with the number of grid points $N = (L/\Delta a)^2$. We believe that it is this poor scaling as to why even a world-leading adhesion simulator like Pastewka [41] abstained from making a direct comparison of approach and retraction of an elastomer from

a randomly rough tip and instead has resorted to Persson's contact-mechanics theory [38] to rationalize the observed compression/decompression hysteresis.

A common solution to reducing the continuum-corrections during the approach is to simulate adhesion directly only during retraction, which, however, requires a contact shape optimization to be done, in particular, during the formation of the contact. Such an approach is relatively "cheap" for bodies of high symmetry, such as bodies of revolution. However, it would be prohibitively expensive when applied to irregular surface structures. Moreover, modeling adhesive interfaces with discontinuous stress-displacement relations could scarcely be applied to time-dependent problems, such as bulk visco-elastic hysteresis, which can add to adhesive losses.

The findings for flat-on-flat geometries also apply to uneven surfaces, where the adhesive energy hysteresis $\gamma_{\text{hys}}$ scales similarly unfavorably with mesh size for Hertzian and randomly rough contacts as for flat-on-flat geometries. In particular, we confirmed that quite large Tabor parameters of $\mu_{\text{T}}$ distinctly exceeding ten, are needed to model the approach curve for an adhesive Hertzian indenter [24], while the retraction curve can be modeled quite accurately with a Tabor parameter as small as $\mu_{\text{T}} = 2$ or even $\mu_{\text{T}} = 1$.

In our simulations, the elastomer's surface facing the indenter was displacement controlled, which is difficult to achieve experimentally due to bulk elasticity and the compliance of the loading apparatus. However, this mode of operation should be seen as a bonus allowing additional insight into the dynamics of adhesion to be gained. For example, a critical (tensile) stress $\sigma_{\text{c}}$ can be determined, at which the elastomer's surface flattens out upon retraction in addition to the maximum tensile stress, or, pull-off stress, which is measured when the retraction is load driven. Moreover, it may be useful to know the tensile stresses in the simulation of adhesive process during detachment processes, since any local grid point in an adhesive simulation is always in between being displacement and load-driven so that knowledge of the tension as a function of separation is often needed even at those separations that would be macroscopically unstable in a load-driven operation.

Our outlook on pull-off forces between randomly rough surfaces suggests that integrating out roughness effects at the small scales reduces not only the adhesion to be used in a cohesive-zone model but also increases the rate of interaction. Both effects combined substantially reduce the stiffness of the contact problem. A true challenge, however, will be to coarse grain the cohesive-zone models so that adhesive hysteresis, including preload effects on the pull-off force, as observed, for example, in structured micro pillars [47], can be modeled. This would allow for the effective simulation of stress-sensitive adhesives.

**Author Contributions:** M.H.M. designed the project. A.W., Y.Z. and M.H.M. performed the simulations and analyzed the data. A.W. and M.H.M. wrote the manuscript. All authors have read and agreed to the published version of the manuscript.

**Funding:** This research was funded by the German Research Foundation (DFG) through grant number MU 1694/5-2.

**Institutional Review Board Statement:** Not applicable.

**Informed Consent Statement:** Not applicable.

**Data Availability Statement:** The data presented in this study are openly available at https://figshare.com/s/ef478970a63423853c18. The uploaded files are xmgrace files, which have ascii format allowing each data set to be extracted from the file with any text editor.

**Conflicts of Interest:** The authors declare no conflict of interest.

## Abbreviations

The following abbreviations are used in this manuscript:

$\Theta(x)$　　heaviside step function
$\alpha$　　relative linear dimension of contact
$\bar{\alpha}$　　relative linear dimension of non contact
$\delta(x)$　　Dirac delta function
$\gamma(z)$　　distance-dependent surface energy
$h_{\text{std}}$　　height standard deviation
$\lambda_{\text{r,s}}$　　wavelength of rolloff and cutoff
$\mu_r ho$　　proportionality coefficient
$\mu_{\text{T}}$　　Tabor parameter
$\rho$　　range of interaction
$\tilde{\rho}$　　range of interaction in units of $\gamma/u_0$
$\sigma_0$　　mean stress
$\sigma(r)$　　stress in real space
$\tilde{\sigma}(q)$　　Fourier transform of the stress
$E^*$　　contact modulus
$F_{\text{pull}}$　　pull-off force
$F_{\text{pull}}^{\text{JKR}}$　　pull-off force in JKR model
$N$　　number of grid points
$L$　　linear system size
$J_\alpha(x)$　　Bessel function of the first kind
$R_{\text{c}}$　　(characteristic) radius of curvature
$V_{\text{ela, int}}$　　elastic or interfacial energy
$\Delta a$　　linear mesh size
$V_{\text{hys}}$　　energy hysteresis
$V_{\text{hys}}^{\text{JKR}}$　　energy hysteresis in JKR model
$\hat{a}$　　relative contact area
$a_{\text{c}}$　　contact radius
$g(\mathbf{r})$　　interfacial separation, or, gap, as function of in-plane coordinate
$n_{x,y}$　　number of unit cells in $x$ or $y$ direction
$p$　　pressure
$p_c$　　critical pressure
$\tilde{p}$　　pressure in unit of $\gamma/u_0$
$q$　　in-plane wave number
$q_{\text{r,s}}$　　roll-off and cut-off wave number
$\mathbf{q}$　　in-plane wave vector
$u_0$　　mean displacement
$\tilde{u}_0$　　mean displacement in units of $\sqrt{\gamma L/E^*}$
$u_g$　　mean gap
$u(r)$　　displacement in real space
$\tilde{u}(q)$　　Fourier transform of the displacement
$v_{\text{ela, int}}$　　elastic or interfacial energy density
$\hat{v}_{\text{ela, int}}$　　elastic or interfacial energy density normalized to contact area

## Appendix A

The analytical treatments of the defect patterns presented in this section are not fully from first principles, i.e., from using solely the stress-displacement relation introduced later in Equation (A5). The spatial stress and displacement profiles observed in the simulations for thin and thick ridges, respectively, enter the calculations. Both profiles turn out to be proportional to $\sqrt{1 - (x/a_{\text{c}})^2}$. Meaningful approximations to this proportionality yield similar results, i.e., functions that are zero for $|x| > a_{\text{c}}$), while symmetric, positive, and continuous otherwise. Numerical constants in the final results change only slightly.

*Appendix A.1. Thin-Ridge Limit*

In this section, we derive an expression for the asymptotic $\hat{v}(\alpha \to 1)$ dependence for thin ridges, starting from the (known or rather observed) stress profile $\sigma(x)$ in a thin ridge. Towards this end, we calculate the mean gap $u_0$ from $\sigma(x)$ and then equate $u_0 \sigma_0/2$ with the work done by the indenter to deform the elastic body.

The stress in the thin ridge satisfies

$$\sigma(x) = \frac{4\sigma_0}{\pi} \sqrt{1 - (x/a_{\rm c})^2}\, \Theta(a_{\rm c} - x). \tag{A1}$$

For the following calculations, we chose the domain such that $0 < 2x \le L$ and placed the ridge symmetrically around $x = 0$, so that only the cosine Fourier transform of stress $\tilde{\sigma}_{\rm c}(q)$ and displacement fields $\tilde{u}_{\rm c}(q)$ are needed. Here, $a_c = \bar{\alpha} L/2$ is the half-length dimension of thin ridge. The following convention for the Fourier transform is used

$$\sigma(x) \;=\; \sum_{n=0,1,\dots} \tilde{\sigma}_{\rm c}\, \cos(q_n\, x) \tag{A2}$$

$$\tilde{\sigma}(q_n) \;=\; \frac{2 - \delta_{n,0}}{L} \int_{-L/2}^{L/2} dx\, \sigma(x)\, \cos(q\, x) \tag{A3}$$

with $q_n = n/(2\pi L)$. Thus,

$$\tilde{\sigma}_{\rm c}(q_n) = 2\,(2 - \delta_{n,0})\, \sigma_0\, \frac{J_1(q_n a_{\rm c})}{q_n a_{\rm c}}. \tag{A4}$$

Using the general relation for the Fourier transforms of stress and displacement,

$$\tilde{\sigma}(q_n) = \frac{q_n\, E^*}{2}\, \tilde{u}(q_n), \tag{A5}$$

which is valid for (frictionless) semi-infinite solids, the mean separation between the two surfaces is obtained as

$$u_0 \;=\; \frac{1}{L} \int_{-L/2}^{L/2} dx \{u(0) - u(x)\} \tag{A6a}$$

$$\;=\; \sum_{n \ne 0} \tilde{u}_{\rm c}(q_n) \tag{A6b}$$

$$\;=\; \frac{8\, a_{\rm c}\, \sigma_0}{E^*} \sum_{n \ne 0} \frac{J_1(q_n a_{\rm c})}{(q_n a_{\rm c})^2} \tag{A6c}$$

$$\;\approx\; \frac{4\sigma_0 L}{\pi E^*} \int_{\pi\bar{\alpha}}^{\infty} dq'\, \frac{J_1(q')}{q'^2} \tag{A6d}$$

$$\;\approx\; \frac{4\sigma_0 L}{\pi E^*} \left\{ -\frac{\ln(\pi\bar{\alpha})}{2} + c \right\} \quad \text{for } \pi\bar{\alpha} \ll 1, \tag{A6e}$$

where the constant $c$ was deduced numerically to be $c \approx 0.30797$. In Equation (A6), $J_1(x)$ denotes a Bessel function of the first kind, for which $J_1(x) \approx x/2$ when $x \ll 1$. This approximation proves useful to determine the prefactor of the $\ln(\pi\bar{\alpha})$ term. Moreover, the $\sum_n f(q_n)$ was approximated with an integral $\frac{L}{2\pi} \int_0^{\infty} dq\, f(q)$.

Equation (A6e) can be solved for $\sigma_0$ so that using $v_{\rm ela} = u_0 \sigma_0/2$

$$v_{\rm ela} = \frac{\pi}{-4\ln(\pi\bar{\alpha}) + 8c}\, \frac{E^* u_0^2}{L} \tag{A7}$$

is obtained.

*Appendix A.2. Thick-Ridge Limit*

For the thick-ridge limit, we proceed similarly as for the thin-ridge limit. However, we now chose the center-of-mass of the non-contact pattern to coincide with $x = 0$ and define $\bar{a}_c$ as half of the non-contact width. Moreover, we now observe the displacement to satisfy

$$u(x) = \frac{4u_0}{\pi} \sqrt{1 - (x/\bar{a}_c)^2}\, \Theta(\bar{a}_c - x). \tag{A8}$$

Using the cosine Fourier transform, the results for the stress obtained in Appendix A.1 can be used again, so that

$$\tilde{u}_c(q_n) = 2\,(2 - \delta_{n,0})\, u_0\, \frac{J_1(q_n a_c)}{q_n a_c}. \tag{A9}$$

Thus, the elastic energy stored in the defect pattern is

$$
\begin{aligned}
v_{\text{ela}} &= \frac{E^*}{4} \sum_{n=1,2,\dots} q_n\, \tilde{u}_c^2(q_n) && \text{(A10a)} \\
&= 4E^* u_0^2 \sum_{n=1,2,\dots} \frac{J_1^2(q_n a_c)}{q_n a_c^2} && \text{(A10b)} \\
&\approx \frac{2\,E^* u_0^2\, L}{\pi\, a_c^2} \int dq'\, \frac{J_1^2(q')}{q'} && \text{(A10c)} \\
&= \frac{E^* u_0^2\, L}{2\,\pi\, a_c^2} && \text{(A10d)}
\end{aligned}
$$

Since $a_c = (1 - \alpha)\, L/2$, it follows that

$$\hat{v}_{\text{ela}} = \frac{2}{\pi(1 - \alpha)^2}. \tag{A11}$$

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
