# Peer review of "Modeling Adhesive Hysteresis"

_lubricants, doi:10.3390/lubricants9020017_

Round 1
Reviewer 1 Report
In this paper, the authors quantify the ensuing unavoidable energy loss for rigid indenters with surface flat and Hertzian surface profiles using analytical and numerical methods. Experimental result reveals that the leading-order corrections of the energy loss being due to finite-range adhesion only disappear with the third root of the linear mesh size, while leading-order errors in the pull-off force disappear linearly. I do have some remarks for improving the paper.
- The authors should illustrate the concept of different types of hysteresis, such as displacement hysteresis and energy hysteresis.
- The main contributions of this paper should be written more clearly.
- The authors should clarify the advantages of the proposed method relative to existing approaches.
- In the process of calculating the total energy (Eq. 15), the authors should further explain the relationship between the total energy and square domain area “A”.
- Please analyze the reason that the numerical results of dimensionless elastic energy ((blue diamonds) evidently diverge from the black line in Fig. 9.
- Please describe the influence of the center-of-mass of the non-contact on the analysis of the thin-ridge limit.
- Throughout the paper, the standard of English writing needs to be improved.
Author Response
We thank you very much for your time and effort to review our manuscript.
Unfortunately, we do not really know how to address your comments.
You want us to explain "the concept of hysteresis". This is standard material, which can be found on Wikipedia and any reasonable physics text book. The area enclosed in a hysteresis loop is the dissipated energy (density). There is no concept of energy hysteresis.
We feel that the abstract is a summary of the main findings. The one thing missing is our suggestion for how to pick the range of adhesion in a simulation. This, however, is the only equation in the conclusions section, which is augmented with a table. We don't know how to make things more clear.
We have no idea how we can help you with your questions 4-6. My Ph.D. student and my co-author were both able to rework through all equations. My Ph.D. student said it was tough to get through all equations and that it sometimes took him some time to see that all information was provided. And my co-author was able to find small typos like missing factors of two. So my impression is that the information is complete, even though it might be hard to digest.
Regarding point 7: I am not a native speaker and I am sure that there are some common mistakes made by Germans in the manuscript. However, I cannot detect them. Despite of this, I feel that my English is not so poor as to make the paper unreadable. It would be insane to request the level of a native speaker by every author.
Reviewer 2 Report
Review of "Modeling adhesive hysteresis" by Wang Zhou and Müser
The authors discuss adhesion instabilities in particular with a view to the design of cohesive-zone models. The paper is rigorous and well written (although not easy to read), however the authors seem to make some confusion, or at least I am confused after trying to read the paper several times. The MS gives the impression to not distinguish sufficiently between true physical phenomena which depend on the cohesive law form, and mesh-dependent error. In various places, it seems that cohesive models can be adjusted to make more efficient modelling.
In general, the entire paper seem to pose the question that the cohesive model "reproduces the (load-approach) relation for zero-range adhesion as accurately as possible for a given, fixed number of grid points". But is this a well-posed question? Given the zero-range adhesion is an idealization, whereas a cohesive model is more realistic, containing more information than just surface energy as in the JKR-Griffith model, why would someone who disposes of a numerical method able to use cohesive element, want to approach the JKR-Griffith solution, which is only a limit which in some cases may not be realistic at all?
The JKR-Griffith idealization holds, as the authors have found themselves, and other before them, only for very large Tabor parameters, i.e. very large spheres, and surely it would not make much sense for a rough surface, where JKR cannot be used at small scales for sure. If one were to use JKR zero-range model for a rough surface, the response would highly depend on the smallest scale we use for the geometry, which is certainly not the correct physical solution. Perhaps the paper is an attempt to generalize the discussion of the method by Popov and others to model the JKR limit using a "mesh-dependent" criterion, whereby the max stress is adapted to the mesh-size. But while the latter method has the clear advantage of simplicity (it is not a full cohesive model like the present authors use), it contains no more than the surface energy as the adhesion parameter, and it would suffer from the same limitations above when extended to anything more than large soft bodies without roughness.
Therefore, I am not really sure what the recipes the authors find are any useful, except for some academic interest. If one disposes of the cohesive model algorithm, one should attempt to input in it the true form of the force-separation law, or at least some reasonable approximation, containing the limit stress and the range of attraction: perhaps the authors could equally study convergence of the solution with respect to an "exact" solution using, say, the Lennard-Jones shape, with some simplified cohesive models: this would make the paper probably more interesting. Would the resulting criteria be related to what they find?
More comments follow:-
∙ In the case of a single asperity, Ciavarella et al (2017) have recently clarified that the jump-in phenomenon and therefore the energy dissipation hysteresis highly depends on the Tabor parameter and this energy converges to the JKR theory only rather slowly for much higher values of Tabor parameter than those for which JKR is normally assumed, like μ>5), and would also depend on the exact form of the cohesive law. So one would need really big spheres to apply the JKR or Griffith analysis, and play with cohesive laws as one like since the results would not depend on them. This slow convergence seems to have been obtained also in this paper and perhaps the earlier result should be mentioned. How these results apply to "asperities" in a rough surface contact is not obvious, as the concept of asperity becomes quite questionable, and asperities tend to be very small anyway.
∙ When an elastomer approaches a flat surface, instabilities are well known to develop and form buldges (see Sarkar et al, Ghatak et al, Joe et al.), and the transition between the flat form , and the buldge does depend on the precise form of the interaction law, as one can determine from a linear stability analysis (and also the loading apparatus stiffness, for that matter).
∙ when we consider a rough contact long wavelength instabilities can also form and give rise to hysteresis (which is not a form of "asperity-scale" hysteresis). In these respects, there is some confusion in the literature, as authors tend to use parameters which are meaningful for Hertzian problems, for the much more complex multiscale roughness. For example, they use the "DMT" regime identification based on the size of the small scale asperities, while it is clear that the instabilities which can form are due to long wavelength and their parametric dependence is not on the local Tabor parameter. Perhaps a more precise definition would be "hysteretic" and "non-hysteretic" regimes, although there could be here also local and more global energy dissipation, so I would be careful. Perhaps the authors should mention the two papers by Joe and Barber and that of Radhakrishnan & Akarapu, where the authors identify regimes of hysteresis. But again, I don't see what the JKR-Griffith approximation could indicate here.
In conclusion, while the paper contains some interesting results, I am not sure if they are well-posed questions or perhaps I misunderstood them entirely. I think a major revision would be beneficial perhaps hoping the above comments could help the authors.
References
Ciavarella, M., Greenwood, J. A., & Barber, J. R. (2017). Effect of Tabor parameter on hysteresis losses during adhesive contact. Journal of the Mechanics and Physics of Solids, 98, 236-244.
J Joe, M Scaraggi, JR Barber Effect of fine-scale roughness on the tractions between contacting bodies Tribology International 111, 52-56
Joe, J., Thouless, M. D., & Barber, J. R. (2018). Effect of roughness on the adhesive tractions between contacting bodies. Journal of the Mechanics and Physics of Solids, 118, 365-373.
A Ghatak, MK Chaudhury Adhesion-induced instability patterns in thin confined elastic film- Langmuir, 2003 - ACS Publications
J Sarkar, V Shenoy, A Sharma Spontaneous surface roughening induced by surface interactions between two compressible elastic films
- Physical Review E, 2003 - APS
Radhakrishnan, H., & Akarapu, S. (2020). Two-dimensional finite element analysis of elastic adhesive contact of a rough surface. Scientific reports, 10(1), 1-9.
Author Response
I thank you very much for your detailed report and apologize that you read it several times and still did not know what to do with it. I perfectly understand your situation. The manuscript evolved over a long time and for a large part it was meant to guide my PDF how to intelligently chose the range of interaction when simulating the dynamics of adhesive processes. I thought that what I teach to him would also helps others. Apparently, I failed miserably, because also my new Ph.D. student who is smart and who read the manuscript keeps having a hard time to make good choices on parameters. I will get back to this issue later.
The manuscript was also a guideline for myself and an excuse to write down (probably re-invent as I confess in the manuscript) linear fracture mechanics. Our field has more and more physicists working. These people, like me, never took a course on elasticity or on solid mechanics. Thus, we have no intuition for terms like energy release rate or stress intensity factor, which any good article in the field assumes to be known. However, like other physicists, I know dimensional analysis, I know how to write down a total energy, and I know how to use the Fourier transform. And these are the only ingredients that I use in the article. Thus getting to results like our relations between dimensionless displacement and dimensionless stress from "first-principles" is a lot easier for me than if I had to use, say, Jim Barber's book on elasticity. I believe that this book is absolutely excellent, and yet, I have a hard time penetrating, as I come from a different background. I would not know how to use this book (even Popov's book which is more physics-like) to derive any result from section 3. So, my derivations might be confusing to someone with a good background in solid mechanics, but it could help physicists. This is why I would like to keep the arduous section 3.
Your report reveals very clearly to me that you understand the topic and that you know the literature. I must confess that I was particularly embarrassed to not have known the first of your references (Ciavareall et al.) because I claimed a result to be new, which had been well established as you rightfully pointed out. I was so embarrassed that I considered to withdraw the manuscript, in particular in light of the fact, that I would not cite Ciavarella for reasons of personal difficulties. While all references were useful (with potentially one exception), this one was particularly important. It is now cited 8 times in the manuscript, including in the introduction, conclusions, analytical section, and the results section.
To address other points, allow me to give a heuristic answer to you, also to abbreviate my response: Pressure-senitive adhesives (that do not form chemical bonds) would probably not be possible without short-range adhesion. We want to simulate them, but I would argue that the new version of the submitted paper is the first one to show clear adhesive hysteresis for a *two-dimensional* interface. And dimensionality matters, as hysteresis is more difficult to obtain in D=2 than in D=1.
Stong adhesive hysteresis is apparently often observed (but rarely published): things often only stick after the elastomer is squeezed hard against a substrate, i.e., the absence of any sign of adhesion on approach and clear adhesion on retraction. Greg Sawyers contribution to the contact-mechanics challenge was one such example. Bo Persson would say that this is viscoelasticity. But I don't believe so. Our new figure 14 (b) looks a lot like what Greg showed to me at the time.
Why has no-one simulated such a curve using a two-dimensional interface? Scaraggi and Carbone cannot do it, because their GFMD is not super efficient and they would have a hard time to equilibrate. I am not sure about their choices of CZMs either. Popov and Li cannot do it, because surfaces only see each other when they touch, so their contact formation is unrealistic or incredibly inefficient (As anyone, who would only assign a maximum tension on retraction). Radhakrishnan used finite elements. His system sizes are the 1/4096 fraction of what we handle. I had a hard time understanding what they use for a CZM and found this to be the one single reference that was not super helpful. Perhaps I would understand it better if I read it as many times as you read my manuscript.
Besides the method itself, I believe that it is the ignorance of how to handle short-range adhesion efficiently that keeps people from running simulations like we now present in our new figure 14.
If others were in a position to run simulations like we present in figure 14, they would have run them. Junki Joe came closest, but his interactions are rather long ranged and thus much easier to handle than the short-range CZMs that we use. Our curve that looks like his can be obtained with systems that are 256 times smaller than those with which we do short-range adhesion.
The last point concerns the readability. You share your difficulty to distinguish between the discussion of true physical phenomena and mesh-size dependent error. I find those difficult to disentangle for the following reason:
If I see an effect that is due to finite-range adhesion than it is a real effect, if adhesion happens to have that range, but it is an artefact if adhesion is short ranged. We now raise a flag, where we emphasize this point.
As a final note, let me say that one reason why I wrote this manuscript is that it was an important step toward the modeling of industry-relevant adhesive devices. We want to explore how the roll-off wavelength might affect the optimum geometry of adhesive devices. What I thought to be an easy exercise turned out to be really difficult. One of the reasons was that the predesigned CZMs did not appear to be helpful.
Round 2
Reviewer 2 Report
The authors have improved the paper significantly. Perhaps the only weak remaining part of the paper is when they discuss Pastewka-Robbins criterion.
In Refs. [1, 2] it is shown that for self-affine fractal surfaces with power-law
PSD and low fractal dimensions (D<2.4), stickiness is magnification-independent, and the PR observation was affected by limitations in the computer capabilities and the use of narrow spectra. This is in agreement with findings of Joe Toughless and Barber. In Ref.[3] it has been shown that for self-affine surfaces with power law PSD and high fractal dimension, PR observation is still wrong, as there is still convergence and even Persson-Tosatti is wrong. The only remaining theory which remains correct
is BAM of Ciavarella, and perhaps Persson-Scaraggi DMT one, while Joe Toughless and Barber confirms this result again.
If the authors make these small modifications, the paper becomes very good, and should be published.
[1] Violano, G., Afferrante, L., Papangelo, A., & Ciavarella, M. (2019). On stickiness of multiscalerandomly rough surfaces. The Journal of Adhesion, 1-19
(12) (PDF) Stickiness of randomly rough surfaces with high fractal dimension: is there a fractal limit?. Available from: https://www.researchgate.net/publication/346969643_Stickiness_of_randomly_rough_surfaces_with_high_fractal_dimension_is_there_a_fractal_limit [accessed Dec 27 2020].
[2] Ciavarella, M. (2020). Universal features in “stickiness” criteria for soft adhesion with roughsurfaces. Tribology International, 146, 106031.
[3] Ciavarella, M. (2020) Stickiness of randomly rough surfaces with high fractal dimension: is there a fractal limit?
https://www.researchgate.net/publication/346969643_Stickiness_of_randomly_rough_surfaces_with_high_fractal_dimension_is_there_a_fractal_limit
Stickiness of randomly rough surfaces with high fractal dimension: is there a fractal limit?. Available from: https://www.researchgate.net/publication/346969643_Stickiness_of_randomly_rough_surfaces_with_high_fractal_dimension_is_there_a_fractal_limit [accessed Dec 27 2020].
Author Response
We thank the reviewer for his positive endorsement of our manuscript and we also thank him once more for showing us a way for how to improve the manuscript with his first report.
With respect to the new report, we feel that an analysis in terms of a BAM, as conducted by Ciavarella, is not meaningful. This is mostly because the contact topography of BAMs is flawed, i.e., way too centered near the highest asperity. In addition, Ciavarella has made so many mistakes when using Persson theory in the past, that I simply lack the time to identify new mistakes in the new works. By the time, I find them, he has published a new preprint and I am back to square zero. Responding to his insane comment on the Contact-Mechanics-Challenge took out 2 months from my research time. Probably he scrabbled it down in an afternoon or two.
On a scientific note, we may note that we have increasing evidence from rigorous simulations (not using the uncontrolled bearing-area approximations that Ciavarella uses and larger systems than in our submitted work) that there is a kink in the curve pull-off force versus surface energy very close to the point where Pastewka and Robbins predicted it to be. This kink can occur at very small pull-off stresses, but we always do see a sudden increase in stickiness at that point.
To yet better address the scepticism by the referee, we added the following sentence in the discussion of the PR criterion:
"Yet, system sizes in this and in former work may be too small to draw valid
conclusions in favor of the PR criterion, while the flawed contact-geometries
in so-called bearing-area models render arguments against it questionable."
We then state that the PR criterion is controversially discussed and cite three high-quality works taking issue with the PR criterion. Of course, we could add a citation to a BAM work. But I don't see that it would add much and only credit somebody who has publicly spewed lies against me by arguing (amongst other things) that I did not allow GW-approaches to participate in the challenge, when indeed, I invited him personally and in very friendly terms to participate. Ciavarella only started to admit some of his lies when I threatened with legal actions. So the reviewer may understand that I had to jump over my own shadow to give Ref. [24] the friendly exposure it has in the new form. And he may understand that I only cite Ciavarella when I am absolutely certain that he makes a valid point for the right reason.